# Differential chondrogenic differentiation between iPSC derived from healthy and OA cartilage is associated with changes in epigenetic regulation and metabolic transcriptomic signatures

Nazir M Khan[1,2], Martha Elena Diaz-Hernandez[1,2], Samir Chihab[1,2], Priyanka Priyadarshani[3], Pallavi Bhattaram[1], Luke J Mortensen[3,4], Rosa M Guzzo[5], Hicham Drissi[1,2]*

[1]Department of Orthopaedics, Emory University, Atlanta, United States; [2]Atlanta VA Medical Center, Decatur, United States; [3]School of Chemical Materials and Biomedical Engineering, University of Georgia, Athens, United States; [4]Regenerative Bioscience Center, E.L. Rhodes Center for ADS, University of Georgia, Athens, United States; [5]Department of Neuroscience, School of Medicine, University of Connecticut Health, Farmington, United States

*For correspondence:
hicham.drissi@emory.edu

Competing interest: The authors declare that no competing interests exist.

**Abstract** Induced pluripotent stem cells (iPSCs) are potential cell sources for regenerative medicine. The iPSCs exhibit a preference for lineage differentiation to the donor cell type indicating the existence of memory of origin. Although the intrinsic effect of the donor cell type on differentiation of iPSCs is well recognized, whether disease-specific factors of donor cells influence the differentiation capacity of iPSC remains unknown. Using viral based reprogramming, we demonstrated the generation of iPSCs from chondrocytes isolated from healthy (AC-iPSCs) and osteoarthritis cartilage (OA-iPSCs). These reprogrammed cells acquired markers of pluripotency and differentiated into uncommitted mesenchymal-like progenitors. Interestingly, AC-iPSCs exhibited enhanced chondrogenic potential as compared OA-iPSCs and showed increased expression of chondrogenic genes. Pan-transcriptome analysis showed that chondrocytes derived from AC-iPSCs were enriched in molecular pathways related to energy metabolism and epigenetic regulation, together with distinct expression signature that distinguishes them from OA-iPSCs. Our molecular tracing data demonstrated that dysregulation of epigenetic and metabolic factors seen in OA chondrocytes relative to healthy chondrocytes persisted following iPSC reprogramming and differentiation toward mesenchymal progenitors. Our results suggest that the epigenetic and metabolic memory of disease may predispose OA-iPSCs for their reduced chondrogenic differentiation and thus regulation at epigenetic and metabolic level may be an effective strategy for controlling the chondrogenic potential of iPSCs.

## Editor's evaluation

This manuscript demonstrates that iPSCs retain the molecular transcriptional signature associated with a healthy or osteoarthritic (OA) state, depending upon the origin of donor cells. Using iPSCs derived from healthy (AC-iPSCs) or OA cartilage, the data show that epigenetic and metabolism-specific transcriptional signals affect the subsequent differentiation of iPSCs to chondrocytes. These findings significantly contribute to the field of regenerative medicine and pave the way to further design new approaches to control and regulate the differentiation of iPSCs to desired cell types.

## Introduction

Osteoarthritis (OA) is an inflammatory joint disease in which catabolic cascade of events results in cartilage destruction leading to severe joint pain (*Lotz and Kraus, 2010*). While non-surgical procedures such as NSAID (Non-steroidal anti-inflammatory drugs) and steroid injections are helpful, the majority of OA cases ultimately undergo joint replacement therapy. The induced pluripotent stem cells (iPSCs) were recently proposed as a promising source to repair cartilage damage (*Lach et al., 2022*; *Lietman, 2016*). While iPSCs are seriously considered as potential cell sources for regenerative medicine, accumulating evidence suggests that iPSCs from different cell sources have distinct molecular and functional properties (*Bar-Nur et al., 2011*; *Kim et al., 2010*; *Lister et al., 2011*; *Ohi et al., 2011*; *Rim et al., 2018*). It has been reported that iPSCs derived from various somatic cell types exhibited a preference for differentiation into their original cell lineages (*Bar-Nur et al., 2011*; *Marinkovic et al., 2016*). Therefore, the effects of the cellular origin of iPSCs on their lineage-specific differentiation capacity is an important consideration for cell replacement therapies, drug screening, or disease modeling.

Several studies have determined that iPSCs retain a memory of their cellular origin due to residual DNA methylation and histone modification patterns at lineage-specific genes. Thus, this residual 'epigenetic memory' has been shown to bias their subsequent differentiation into their parental/donor cell lineage (*Kim et al., 2011*; *Choompoo et al., 2021*; *Wang et al., 2018*). Although it is known that cellular origin of iPSCs influences their differentiation capacity, the contribution of disease-specific factors on the capacity of iPSC for chondrogenic differentiation remains unknown. Examining potential differences between cells that reside in healthy vs. OA environments, would provide unique insight into the chondrogenic potential of these cells, and their utility in disease modeling. Since OA articular chondrocytes exhibit different features from healthy articular chondrocytes, we posit that the iPSCs derived from these cell states represent the feature of their physiological origin. Thus, the memory of the cells is not only specific to the tissue of origin but also to the physiological status which further influences the differentiation capacity and ultimately the efficiency of tissue regeneration.

In the present study, we aimed to determine whether iPSCs derived from healthy and diseased (OA) cartilage possess differential chondrogenic potential, and whether OA disease status significantly limits their differentiation capacity. To this end, we derived iPSCs from healthy (AC-iPSCs) and OA chondrocytes (OA-iPSCs) and compared their differentiation capacity into chondroprogenitors and chondrocytes. Our results showed that iPSCs derived from healthy chondrocytes (AC-iPSCs) exhibited an enhanced potential for chondrocyte differentiation as compared to OA-iPSCs. Our data further demonstrate that although reprogramming of OA chondrocytes induced pluripotency, the OA-iPSCs retained the changes in epigenetic and metabolic factors associated with pathological conditions of diseased chondrocytes. Retention of this cellular memory may influence their chondrogenic commitment, and thus regenerative capacity for the cartilage repair. Our findings indicate that regulating the epigenetic modifiers and energy metabolism may be an effective strategy for enhancing the chondrogenic potential of iPSCs derived from chondrocytes.

## Results

### Characterization of iPSCs generated from healthy and OA articular chondrocytes

We previously reported the generation of iPSCs from healthy articular chondrocytes (AC-iPSCs) and performed molecular, cytochemical, and cytogenic analyses to determine the pluripotency of generated iPSCs (*Guzzo et al., 2014*). In the present study, we used multiple clones of the previously generated AC-iPSCs (clones #7, #14, and # 15), and compared their pluripotency, progenitor properties, and chondrogenic potential to that of newly generated OA-derived iPSCs (OA-iPSCs) (clones #2, #5, and #8) (*Figure 1A*). These colonies showed positive alkaline phosphatase (ALP) staining, indicating an undifferentiated pluripotent stem cell phenotype of both AC- and OA-iPSC clones (*Figure 1B*). Stemness characteristics of these iPSC clones were evaluated via qPCR assessment of key pluripotency marker genes. The mRNA copy number of *SOX2*, *OCT4*, *NANOG*, and *KLF4* was comparable in AC- and OA-iPSCs (*Figure 1C*) indicating a similar level of stemness identity between these iPSCs. Interestingly, *KLF4* expression was low as compared to the other pluripotency gene in both iPSCs (*Figure 1C*). Pluripotency was also confirmed using immunofluorescence staining and our results

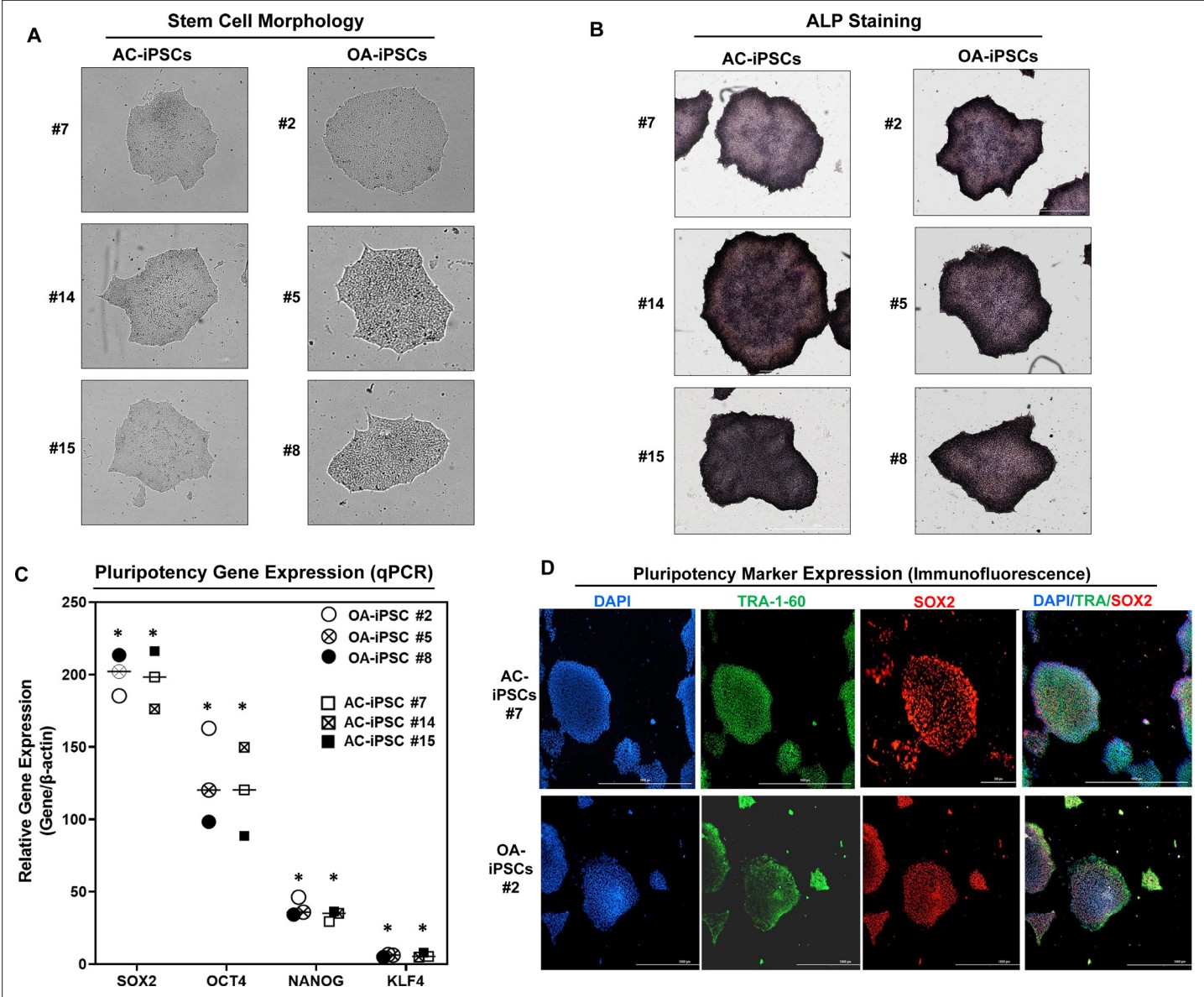

**Figure 1.** Characterization of induced pluripotent stem cells (iPSCs) generated from healthy and osteoarthritis (OA) articular chondrocytes. (**A**) Morphology of the AC-iPSC (#7, #14, and #15) and OA-iPSC (#2, #5, and #8) colonies in monolayer culture on a 0.1% Geltrex-coated plate. Image displays representative experiment (*n* = 3). (**B**) Alkaline phosphatase (ALP) staining of iPSC colonies showing undifferentiated pluripotent stage. Image displays representative experiment (*n* = 3). (**C**) Pluripotency for iPSC colonies showing expression of stemness genes. RT-qPCR (Real time-quantitative PCR) analyses showed induced expression of canonical stemness genes *SOX2*, *OCT4*, *NANOG*, and *KLF4* in AC- and OA-iPSC colonies. β-Actin served as the housekeeping gene and internal control. Results from one representative experiment (*n* = 3). Represented gene expression data are relative to mesenchymal stem-like cells (MSCs) derived from respective iPSC cells. *p ≤ 0.01, as compared to their respective MSCs. (**D**) Immunofluorescence staining of pluripotency markers in AC-iPSCs (#7) and OA-iPSCs (#2) showed expression of surface TRA-1–60 and SSEA-4 antigens in these colonies. DAPI (4′,6-diamidino-2-phenylindole) is used as nuclear counterstain showing blue nuclei. Scale bar, 100 μm. Image displays representative experiment (*n* = 3).

The online version of this article includes the following source data for figure 1:

**Source data 1.** Depicting original raw data related to *Figure 1*.

demonstrated that cell colonies from both AC- and OA-iPSCs showed positive expression of SOX2 and TRA-1–60 proteins (*Figure 1D*).

## MSCs differentiated from AC- and OA-iPSCs exhibit comparable phenotypic features in vitro

Differentiation of human iPSCs into mature chondrocytes requires derivation of an intermediate mesenchymal-like progenitors stage (*Guzzo et al., 2014*; *Guzzo and Drissi, 2015*; *Guzzo et al., 2013*; *Drissi et al., 2015*). Therefore, we generated mesenchymal progenitor intermediate from all three clones of both AC- and OA-iPSCs using our established direct plating method in the presence of serum and human recombinant bFGF (*Guzzo et al., 2013*; *Drissi et al., 2015*). Mesenchymal stem-like cells (MSCs) derived from both AC-iPSCs (termed as AC-iMSCs) and OA-iPSCs (termed as OA-iMSCs) displayed similar phenotypic characteristics of spindle-shaped and elongated morphology (*Figure 2A*). We next performed detailed characterization of iMSCs from both sources to determine their mesenchymal properties. Profiling by qPCR showed significant suppression of stemness genes including *SOX2* and *OCT4*, in both AC- and OA-iMSCs as compared to the parental undifferentiated AC- and OA-iPSCs, respectively (*Figure 2B*). We also analyzed the expression of marker genes associated with the mesenchymal lineage and our results showed that mRNA expression of *TWIST1* (an epithelial-to-mesenchymal transition-related gene), *COL1A1* (an ECM (extracellular matrix) molecule synthesized by MSCs), and *RUNX1* (a transcription factor expressed in mesenchymal progenitors) was significantly higher in both iMSCs as compared to the pluripotent parental iPSCs (*Figure 2B*).

Consistent with the standard criteria defined by the International Society of Cell and Gene Therapy (ISCT) (*Dominici et al., 2006*), immunophenotypic analyses revealed cell surface expression of all typical MSC markers in both iMSC progenitors with high enrichment of CD44, CD73, CD90, CD105, and CD166 (*Figure 2C*). Conversely, both iMSCs largely lacked expression of the definitive hematopoietic lineage marker CD45, and the endothelial marker CD31. Comparative analysis of these markers in AC- and OA-iMSCs showed comparable expression levels suggesting an identical immunophenotype of both iMSCs (*Figure 2D*). To determine the multipotential of these iMSCs, we performed their trilineage differentiation using in vitro adipogenic, osteogenic, and chondrogenic differentiation assays (*Figure 2—figure supplement 1A–C*). Although both MSCs could clearly form osteoblasts, adipocytes, and chondrocytes, AC-iMSCs displayed enhanced chondrogenic potential as evidenced by increased deposition of Alcian blue positive extracellular matrix compared to OA-iMSCs (*Figure 2—figure supplement 1B*). We also performed gene expression analysis for the markers of osteoblast differentiation in AC- vs. OA-iMSCs over a time course of differentiation process (days 7, 14, and 21). Our qPCR analyses revealed that transcripts levels of osteogenic genes such *RUNX2*, *OSX*, and *COL1A1* was higher at 7 days in comparison to undifferentiated iMSCs levels (day 0) (*Figure 2—figure supplement 1C*). The transcript level of these genes was further increased at days 14 and 21 of osteogenic differentiation. Interestingly, the expression of these osteogenic genes in AC-iMSCs was not statistically significant when compared to OA-iMSCs at all time points analyzed (*Figure 2—figure supplement 1C*).

This finding is consistent with the Alizarin red staining showing that AC- and OA-iMSCs exhibit in vitro differentiation assays for adipocytes and osteoblasts. However, AC-iMSCs similarities in osteogenic differentiation. Altogether, the data suggested that iMSCs derived from AC- and OA-iPSCs exhibit similarities in morphology, immunophenotype, and multipotency as evidenced by as evidenced by in vitro differentiation assays for adipocytes and osteoblasts. However, AC-iMSCs displayed increased chondrogenic differentiation as compared to OA-iMSCs.

## AC-iMSCs exhibit enhanced chondrogenic potential in vitro

We next evaluated whether AC-iMSCs exhibit higher propensity for chondrogenic differentiation as compared to OA-iMSCs. Chondrogenic differentiation of these iMSCs was examined using our well-established pellet culture method using chondrogenic media in the presence of human recombinant BMP-2 (*Figure 3A*; *Guzzo et al., 2014*; *Guzzo and Drissi, 2015*; *Guzzo et al., 2013*; *Drissi et al., 2015*). Quantitative PCR analyses of key chondrogenic genes were used to evaluate the potential of AC- and OA-iMSCs to produce chondrocytes at days 7, 14, and 21. When compared to the undifferentiated MSC culture (day 0), induction of *SOX9*, *COL2A1*, *ACAN*, and *PRG4* transcript was significantly increased at day 7, and to a greater extent at day 14 (*Figure 3C*). Interestingly, mRNA expression of

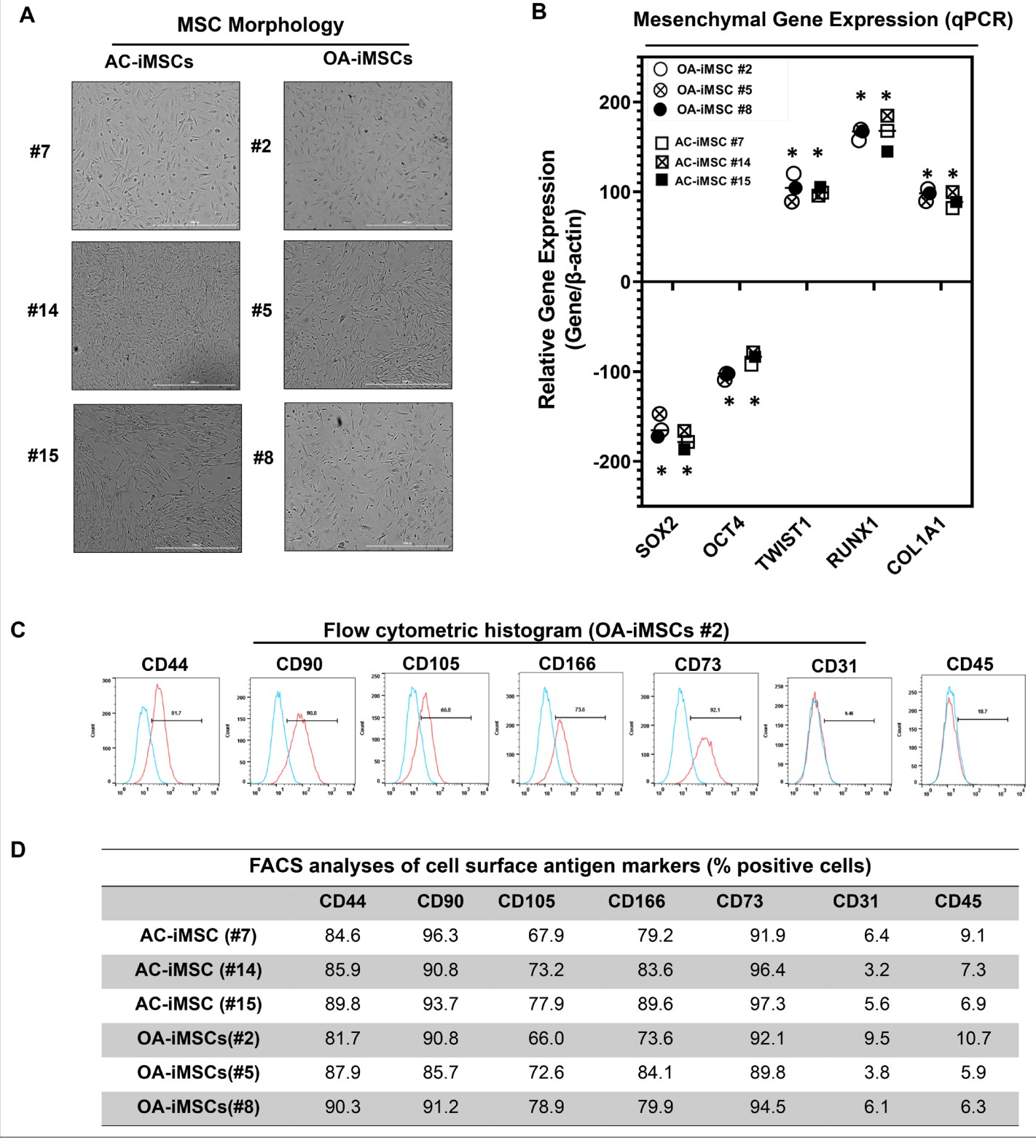

**Figure 2.** Derivation of induced pluripotent stem cell (iPSC)-MSCs (iMSCs) like cells from AC- and OA-iPSCs and characterization of their mesenchymal feature. (**A**) The morphology of the iMSC-like cells (iPSC–MSC) derived from AC- and OA-iPSC showing elongated spindle-shaped cells. Representative images are shown for iMSCs at passages 5–8. Scale bar, 100 µm. Image displays representative experiment (*n* = 3). (**B**) Gene expression analyses by qPCR showing significant suppression of pluripotent markers OCT4 and SOX2, and induction of mesenchymal genes TWIST1, COL1A1, and RUNX1 in the AC- and OA-iMSCs relative to their parental iPSCs. β-Actin served as the housekeeping gene and internal control. Results from one representative

*Figure 2 continued on next page*

*Figure 2 continued*

experiment (*n* = 3). Expression data are represented as fold change relative to respective parental iPSCs. *p ≤ 0.01, as compared to their respective iPSCs. (**C**) Expression of surface antigens in AC- and OA-iMSCs by flow analysis. Representative flow cytometric histogram showing OA-iMSCs (#2) express markers associated with the mesenchymal phenotype (positive for CD44, CD73, CD90, CD105, and CD166; negative for CD31 and CD45) (*n* = 3). Red histogram shows antibody-stained population; blue profile shows negative isotype-stained population. (**D**) Comparative flow cytometry analyses of AC-iMSCs (#7, #14, and #15) and the OA-iMSCs (#2, #5, and #8) showing similar cell surface expression profiles. Results from one representative experiment (*n* = 3).

The online version of this article includes the following source data and figure supplement(s) for figure 2:

**Source data 1.** Depicting original raw data related to *Figure 2B*.

**Source data 2.** Table providing data related to *Figure 2D*.

**Figure supplement 1.** Trilineage differentiation of AC- and OA-iMSCs into osteoblasts, chondrocytes and adipocytes.

*SOX9*, *COL2A1*, *ACAN*, and *PRG4* was significantly higher in AC-iMSCs as compared to OA-iMSCs at all time points analyzed (days 7, 14, and 21) suggesting that iPSC derived from healthy chondrocytes have a significantly higher chondrogenic potential as compared to OA-iPSC (*Figure 3C*).

We also performed chondrogenic differentiation of these iMSCs using high-density adherent micromass culture method. 3D-micromass culture of pluripotent stem cells resembles the formation of prechondrogenic mesenchymal condensations and their differentiation into the chondrogenic lineage (*Guzzo et al., 2013*; *Daniels et al., 1996*). Alcian blue staining of Day 21 micromass culture of AC-iMSCs showed densely stained central core surrounded by a diffusely stained outer cellular layer showing increased accumulation of glycoprotein-rich matrix as compared to OA-iMSCs (*Figure 3B*). Additionally, Alcian blue staining in AC-iMSCs further showed increased cellular outgrowths and

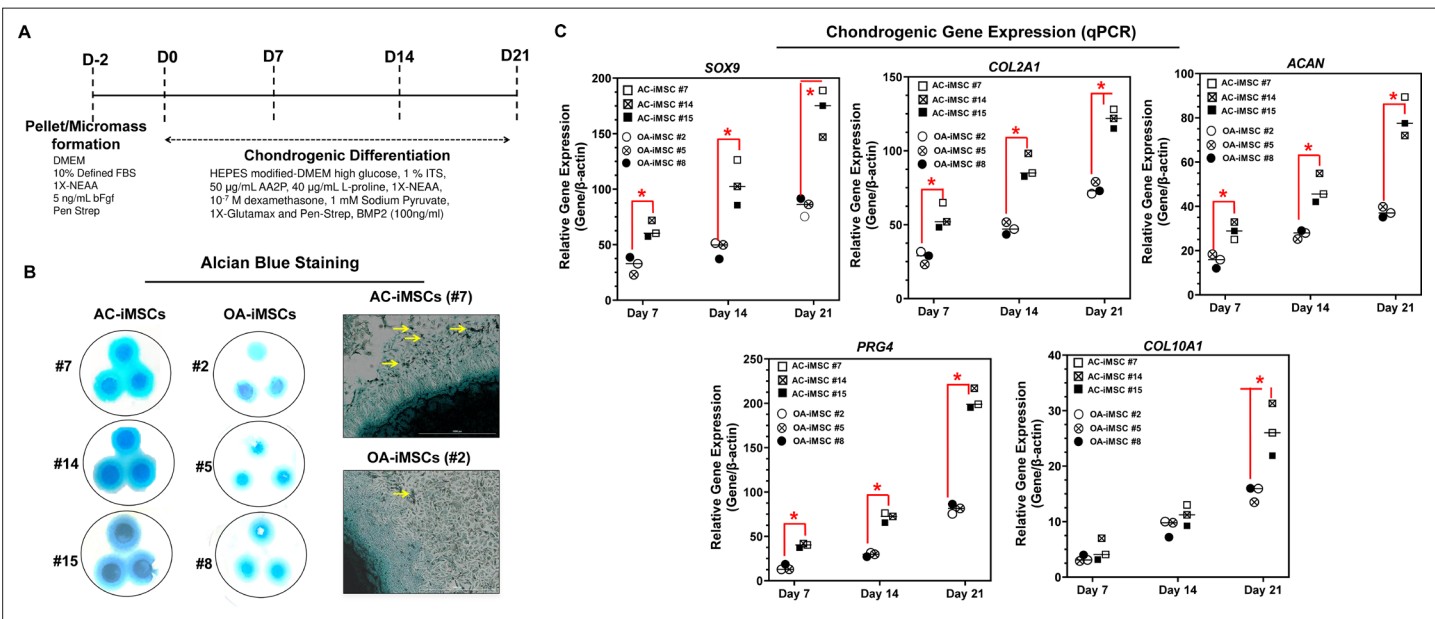

**Figure 3.** AC-iMSCs exhibit superior chondrogenic potential in vitro. (**A**) Schematic showing treatment conditions for in vitro chondrogenic differentiation of AC- and OA-iMSCs using pellet and micromass method. (**B**) Chondrocyte differentiation was shown by Alcian blue staining of micromass cultures in serum-free chondrogenic media for 21 days with 100 ng/ml BMP2. Alcian blue staining revealed accumulation of sulfated proteoglycans indicating enhanced secretion of matrix in AC-iMSC as compared to OA-iMSCs micromass culture. High-magnification images further demonstrated enhanced cellular compaction (yellow arrow) in AC-iMSCs micromass indicating the development of cartilaginous nodules. Scale bar, 100 μm. Image displays representative experiment (*n* = 3). (**C**) Quantitative PCR analyses of the relative transcript levels of chondrogenic genes *SOX9*, *COL2A1*, *ACAN* and hypertrophic gene *COL10A1* in days 7, 14, and 21 pellet culture of all three clones of AC- and OA-iMSCs. β-Actin served as the housekeeping gene and internal control. Values represent fold induction (mean ± standard deviation [SD]) relative to control iMSCs (day 0) from three replicate. *p ≤ 0.01 indicate values are statistically different in OA-iMSCs as compared to their AC-iMSCs at each time point. Results from one representative experiment (*n* = 4).

The online version of this article includes the following source data for figure 3:

**Source data 1.** Depicting original raw data related to *Figure 3*.

cartilaginous nodules, confirming enhanced chondrogenic potential of AC-iMSCs as compared to OA-iMSCs (*Figure 3B*). These Alcian blue staining showing ECM synthesis are in line with the expression data for the matrix genes. These data further indicate that iMSCs derived from OA chondrocytes showed reduced ECM generation upon chondrogenic differentiation which may be a retention of OA phenotype of original cell source.

## AC-iMSCs exhibit distinct transcriptomic signature during chondrogenic differentiation

To examine the underlying transcriptional programs associated with enhanced chondrogenic potential of AC-iMSCs as compared to OA-iMSCs, we performed RNA-seq analysis. We identified gene expression changes at pan-genome levels in day 21 differentiated chondrocytes from AC-iMSCs (#7) and OA-iMSCs (#5). The volcano plot showed that global gene expression profiles of the chondrocytes at day 21 chondrogenic culture of AC-iMSCs were significantly different from the OA-iMSCs (*Figure 4A*). This analysis identified 146 genes that were upregulated, and 263 genes that were downregulated in chondrocytes derived from AC-iMSCs (termed as AC-iChondrocytes) as compared to OA-iMSCs (termed as OA-iChondrocytes) (*Figure 4A*). To validate these findings, we performed quantitative gene expression analysis of a subset of differentially expressed genes (DEGs) such as *FOXS1*, *ADAM12*, *COL1A1*, *COL3A1*, *MATN4*, and *MARK1* during chondrogenic differentiation and analysis confirmed differential expression levels in AC- vs. OA-iChondrocytes (*Figure 4—figure supplement 1*). Additionally, principal component analyses placed AC- and OA-iChondrocytes in two distinct clusters suggesting that chondrocytes derived from AC-iMSCs were genomically distinct from OA-iMSC-derived chondrocytes (*Figure 4B*).

We next performed functional annotation analyses of these differentially regulated genes to determine the enrichment of GO terms and molecular pathways. Our GO analyses demonstrated significant enrichment of several biological processes in AC-iChondrocytes including histone modification, chromatin organization, oxidative phosphorylation, glucose metabolism, chondrocyte differentiation, and ECM organization (*Figure 4C*). These results suggest that pathways related to 'Energy Metabolism' and 'Epigenetic Regulation' play an important role in chondrogenic differentiation of AC-iMSCs. We also performed KEGG pathway analysis and results showed that 'Metabolic Pathways', 'Epigenetic Regulation', and 'Chromatin Organization' are the most enriched pathways in AC-iMSCs (*Figure 4D*). These data suggest that a large proportion of DEGs between AC- and OA-iChondrocytes were involved in 'Energy Metabolic pathways' such as oxidative phosphorylation, glucose metabolism, and protein phosphorylation and 'Epigenetic Regulatory pathways' such as chromatin organization and histone modification. The regulatory genes involved in these pathways such as *HDAC10/11*, *PRMT6*, *PRR14*, *ATF2*, *SS18L1*, *JDP2*, *RUVBL1/2*, *OGDHL*, *ALDH2*, *GCLC*, *GOT1*, *HIF1A*, *COX5A*, and *TRAF6* may create a distinct metabolic and chromatin state in AC-iMSCs which favors its enhanced chondrogenic differentiation.

## AC-iMSCs revealed enrichment of interaction networks related to energy metabolism and epigenetic regulation during chondrogenic differentiation

To determine the functional relationships among genes that were differentially regulated during chondrogenic differentiation of AC- and OA-iMSCs, we performed interaction network analyses. Our analysis identified two major subnetworks distributed in two distinct clusters belonging to energy metabolism and epigenetic regulation suggesting a role for these pathways in chondrogenic differentiation of AC-iMSCs (*Figure 5A, B*). The ClueGO analysis in metabolic gene network cluster showed enrichment of several energy metabolic pathways such as glycolysis, amino acids synthesis, autophagy and biosynthesis, and anabolic pathways suggesting that multiple metabolic signaling networks in energy metabolism may contribute to enhanced chondrogenic potential of AC-iMSCs (*Figure 5A*). Similarly, Epigenetic regulator gene network cluster comprise of several pathways related to histone modification, chromatin regulation, histone acetylation, and chromatin assembly/disassembly. These data suggest that during chondrogenic differentiation of AC-iMSCs, the expressions of several chromatin modifiers were increased, which may regulate key genes involved chondrogenic differentiation (*Figure 5B*).

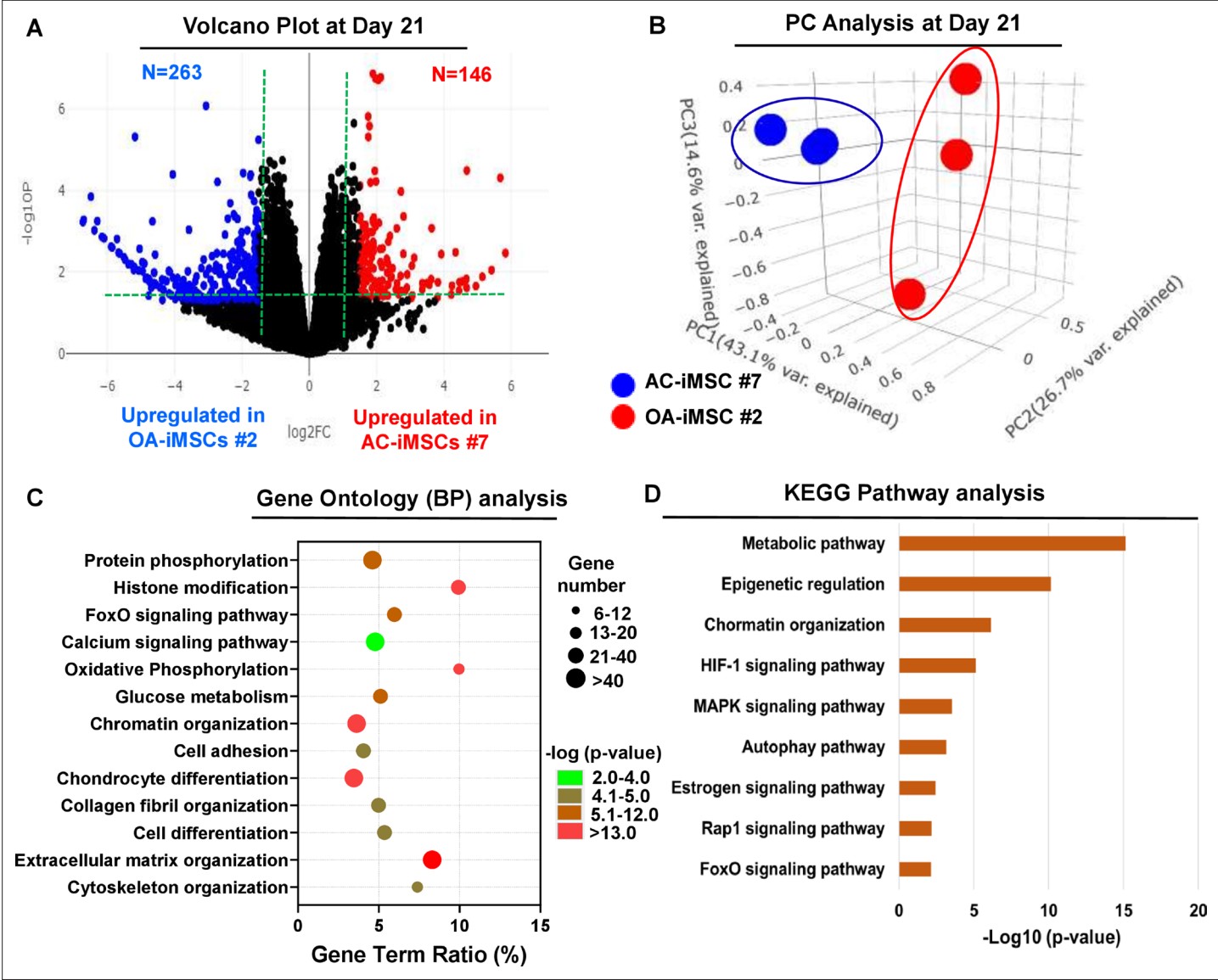

**Figure 4.** AC-iMSCs during chondrogenic differentiation exhibit distinct transcriptome signature. Bulk RNA-sequencing was performed during chondrogenic differentiation of AC- vs. OA-iMSCs and differential gene expression analyses revealing distinct transcriptomic signature. (**A**) Genes with differential expression levels greater than twofold (false discovery rate [FDR] p value <0.05) were visualized as volcano plot showing differential expression of 406 genes. (**B**) Principal component analysis (PCA) using differentially expressed genes (DEGs) showing segregation of AC- vs. OA-iChondrocytes generated during day 21 chondrogenic differentiation using pellet culture. (**C**) Functional annotation clustering using GO analysis for biological process (BP) using DEGs at day 21 chondrogenic differentiation of AC- vs. OA-iMSCs. *Y*-axis label represents pathway, and *X*-axis label represents gene term ratio (gene term ratio = gene numbers annotated in this pathway term/all gene numbers annotated in this pathway term). Size of the bubble represents the number of genes enriched in the GO terms, and color showed the FDR p value of GO terms. (**D**) KEGG (Kyoto Encyclopedia of Genes and Genomes) pathway analysis showing enrichment of molecular pathways contributing to differential chondrogenic potential of AC- vs. OA-iMSCs.

The online version of this article includes the following source data and figure supplement(s) for figure 4:

**Source data 1.** Depicting original raw data related to *Figure 4*.

**Source data 2.** Depicting original raw data related to *Figure 4*.

**Figure supplement 1.** Validation of RNA-seq results by quantitative PCR.

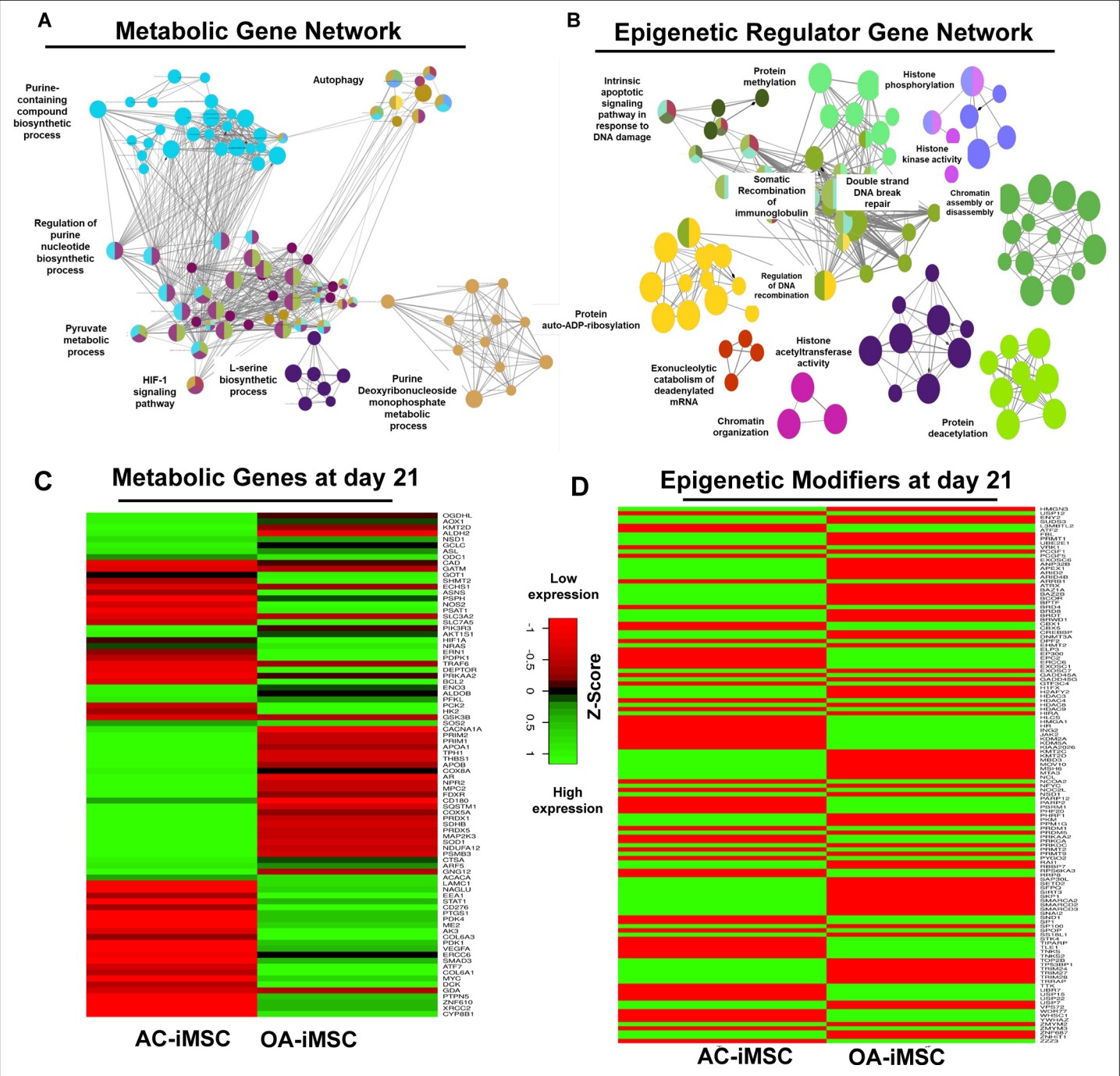

**Figure 5.** Enrichment of metabolic and epigenetic regulators interaction networks during chondrogenic differentiation of AC-iMSCs. (**A, B**) Interaction network analysis using differentially expressed genes (DEGs) at day 21 chondrogenic differentiation of AC- and OA-iMSCs. PPI network of DEGs in AC-iChondrocytes was constructed using STRING database and visualized by Cytoscape. Pathway enrichment analysis in the interaction network was performed using ClueGO analysis which showed enrichment of pathways related to (**A**) metabolic genes and (**B**) epigenetic modifiers. Multiple nodes of metabolic and epigenetic regulators were enriched in these interaction networks suggesting the role of these pathways in differential chondrogenic potential. (**C, D**) Differential expression analyses of the genes involved in these enriched pathways related to energy metabolism and epigenetic regulators. The gene expression was visualized using heatmap analysis for DEGs related to (**C**) energy metabolism and (**D**) epigenetic regulators. Expression values for each gene (row) were normalized across all samples (columns) by Z-score. Color key indicates the intensity associated with normalized expression values. Green shades indicate higher expression and red shades indicate lower expression.

The online version of this article includes the following source data for figure 5:

**Source data 1.** Depicting original raw data related to *Figure 5*.

*Figure 5 continued on next page*

*Figure 5 continued*

**Source data 2.** Depicting original raw data related to *Figure 5*.

**Source data 3.** Depicting original raw data related to *Figure 5*.

To further implicate the role of 'energy metabolism' and 'epigenetic regulator pathways' in differential chondrogenic potential, we analyzed the expression profile of genes involved in these pathways during chondrogenic differentiation of AC- and OA-iMSCs. The heatmap analysis during terminal chondrogenic differentiation (day 21) showed that the expression profile of various metabolic and epigenetic regulator genes exhibits differential expression in AC- vs. OA-iChondrocytes (*Figure 5C, D*). Moreover, the expression profile for metabolic and epigenetic factor genes correlates well with chondrogenic differentiation of these iMSCs further suggesting the importance of these pathways in enhanced chondrogenic potential of AC-iMSCs. Altogether, these data suggest that metabolic and epigenetic regulatory pathways play a role in chondrogenic potential of AC-iMSCs.

## AC-iMSCs at the undifferentiated state showed distinct expression of genes involved in energy metabolism and epigenetic regulation

The data in *Figure 5C, D* show that during chondrogenic differentiation, AC-iMSCs exhibit differential expression for the metabolic and chromatin regulator genes. We next examined whether this differential gene expression profile was intrinsic to AC-iMSCs or acquired during the process of chondrogenic differentiation. To this end, we performed transcriptomic analyses at various stages of chondrogenic differentiation of AC- and OA-iMSCs. Volcano plot analysis identified that AC- and OA-iMSCs at undifferentiated steady state (day 0) exhibited differential expression at pan-genome level with >800 DEGs (*Figure 6A*). We next focused our analysis on the expression of genes involved in energy metabolic and epigenetic regulator pathways. Similar to the level observed at terminal differentiation stage, our analysis revealed that metabolic and chromatin regulator genes also showed significant differences between both cell types at the uncommitted mesenchymal state (*Figure 6B, C*). When compared to OA-iMSCs, the AC-iMSCs expressed higher levels of several metabolic gene involved in glycolysis, amino acid synthesis, autophagy, and anabolic pathways such as *ALDOB*, *CD180*, *SQSTM1*, *ENO3*, *AOX1*, *KMT2D*, *COX5A*, *PRDX1*, *SDHB*, and *ALDH2* (*Figure 6B*). Moreover, differential expression of multiple chromatin modifiers including histone modifiers (eraser, reader, and writers) and chromatin remodeling factors such as *JDP2*, *RUVBL1/2*, *MYBBP1A*, *HDAC10*, *HDAC11*, *USP12*, *L3MBTL2*, and *MUM1* was also observed (*Figure 6C*). Further, differential expression patterns of several epigenetic modifiers at the MSC stage (day 0) were retained at the chondrocyte stage (day 21). For example, *ARID4B*, *BRD4*, *HDAC4*, *HDAC9*, *KDM5A*, and *KMT2C* showed differential expression between OA- and AC-iMSCs at both days 0 and 21 stage of chondrogenic differentiation. These results suggest that differential expression of genes associated with energy metabolism and epigenetic regulation between healthy and OA conditions first occurs at the MSC stage, prior to their overt differentiation to the chondrogenic lineage. Thus, differences in the chondrogenic potential of AC- vs. OA-iMSCs may be associated with differences in expression of metabolic and chromatin modifier genes which influence the chondrogenic capacity of these MSCs.

## Genetically distinct characteristic of AC- and OA-iMSCs was imprint of original cell sources from healthy and OA chondrocytes

Our data as above (*Figure 2A–E* and *Figure 6A–C*) indicated that although AC- and OA-iMSCs exhibit similar morphologic and immunophenotypic characteristics, they are genetically distinct populations that displayed varying efficiencies for chondrogenic differentiation. We therefore postulated that differences in the metabolic and chromatin modifier gene expression patterns observed in OA-iMSCs as compared to AC-iMSCs are attributed to their initial disease status. To explore this, we analyzed the expression profiles of metabolic and chromatin modifier genes from multiple sources of healthy and OA chondrocytes. Thus, we analyzed publicly available RNA-seq data performed on healthy and OA cartilage tissues (GSE114007) (*Fisch et al., 2018*). This analysis revealed that the expression profiles of a large number of metabolic and chromatin modifier genes are differentially expressed in human AC- vs. OA-cartilages (*Figure 7A, B*). Similarly, we detected differential expressions of key metabolic and epigenetic modifiers in our unbiased datasets from AC- vs. OA-iMSCs (uncommitted stage, day

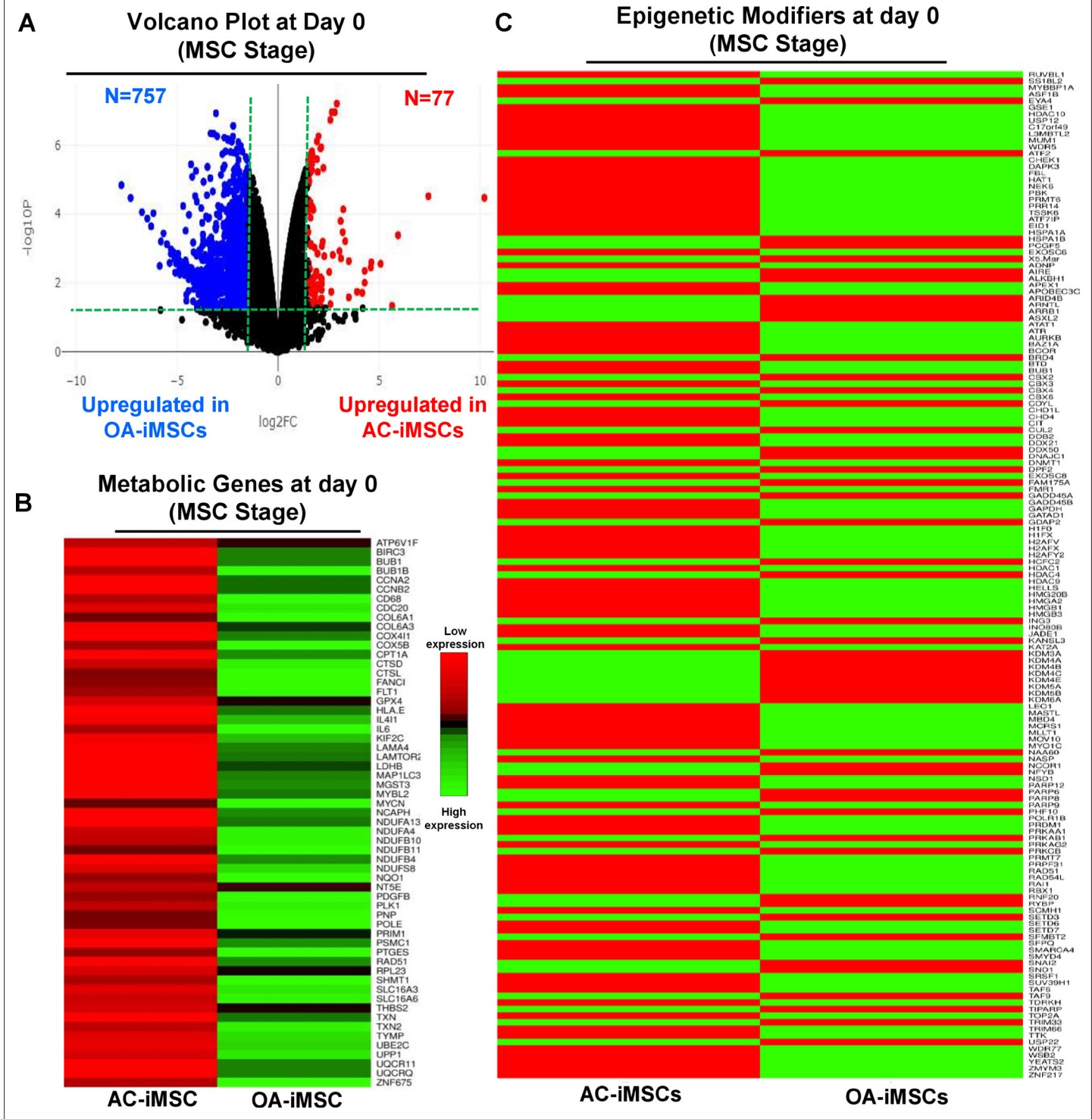

**Figure 6.** AC-iMSCs at undifferentiated state showed distinct expression of genes involved in metabolic and epigenetic regulators. (**A**) Differential gene expression analyses of AC- and OA-iMSCs at day 0 (start of chondrogenic differentiation) showing distinct transcriptomic signature. Genes with differential expression levels greater than twofold (false discovery rate [FDR] p value <0.05) were visualized as volcano plot showing differential expression of 834 genes. (**B, C**) Pathway analysis was performed in 834 differentially expressed genes (DEGs) to show the enrichment of pathways related to metabolism and epigenetic modifiers. Heatmap was used to show expression of the genes related to (**B**) energy metabolism and (**C**) epigenetic regulators. Expression values for each gene (row) were normalized across all samples (columns) by Z-score. Color key indicates the intensity associated with normalized expression values. Green shades indicate higher expression and red shades indicate lower expression.

The online version of this article includes the following source data for figure 6:

*Figure 6 continued on next page*

*Figure 6 continued*

**Source data 1.** Depicting original raw data related to *Figure 6*.

**Source data 2.** Depicting original raw data related to *Figure 6*.

0), suggestive of the persistence of a cellular memory of disease even after reprogramming. Several chromatin modifiers such as *HAT1*, *HDAC10*, *HDAC11*, *PRMT6*, *JDP2*, *ATF2*, *ATF7*, and *WDR5* which showed differentiation expression in healthy and OA cartilage also showed retention of differential pattern in OA- vs. AC-iMSCs. Additionally, AC- and OA-iMSCs to hostile or pathogenic inflammatory environment showed expression of inflammatory genes, however, extent of these pro-inflammatory

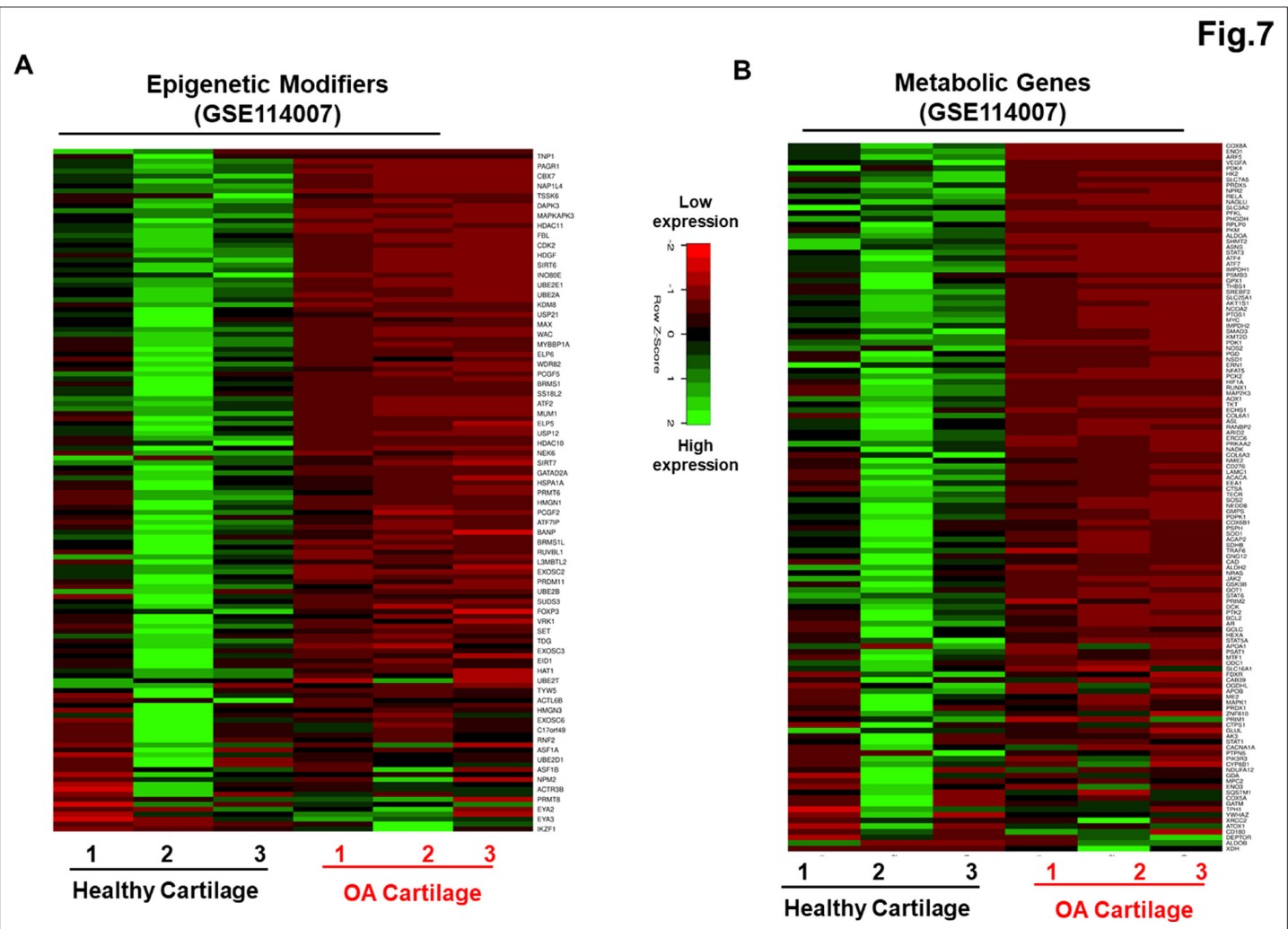

**Figure 7.** Genetically distinct characteristic of AC- and osteoarthritis (OA)-iMSCs was imprint of original cell sources from healthy and OA chondrocytes. The expression of genes involved in energy metabolism and epigenetic modifiers was performed by analysis of publicly available RNA-seq data performed on healthy and OA cartilage tissues (GSE114007). Gene expression was visualized by heatmap analysis for (**A**) epigenetic modifiers and (**B**) metabolic genes. Expression values for each gene (row) were normalized across all samples (columns) by Z-score. Color key indicates the intensity associated with normalized expression values. Green shades indicate higher expression and red shades indicate lower expression.

The online version of this article includes the following source data and figure supplement(s) for figure 7:

**Source data 1.** Depicting original raw data related to *Figure 7*.

**Source data 2.** Depicting original raw data related to *Figure 7*.

**Figure supplement 1.** Inflammatory response of AC- and OA-iMSCs in response to stimulation of IL1α (10 ng/ml).

**Figure supplement 2.** Upstream transcription factor analysis among the epigenetic modifiers between AC- and osteoarthritis (OA)-iMSCs identified the enrichment of several transcription factors including FOXM1 which is predicted to regulate the response of iMSCs in inflammatory environment.

gene expression such as *CCL2*, *CCL3*, *CXCL3*, and *NOS2* was significantly higher in OA-iMSCs as compared to AC-iMSCs indicating that OA-iPSCs retain memory of disease from the tissue of origin derived from OA cartilage (*Figure 7—figure supplement 1*).

Further evaluation of the epigenetic landscapes between OA- and AC-iPSCs may reveal specific changes in the methylome or histone modifications that can be targeted to correct the skewing of OA-iMSCs to provide an equal chondrogenic or immunomodulatory potentials to that of healthy cartilage-derived iMSCs. Among various epigenetic marks expressed in AC- vs. OA-iMSCs, the upstream transcription factor analysis identified several candidate regulators such as FOXM1, IRF3, FOXP1, and MYBL2 (*Figure 7—figure supplement 2*). Together our data suggest that a retained memory of disease during stem cell reprogramming affected the chondrogenic differentiation potential of OA-iMSCs.

## Discussion

iPSCs are viewed as promising cell-based therapeutics for the repair of tissues lacking intrinsic regenerative capacity, including articular cartilage. Multiple studies have cautioned that safe and effective application of iPSCs based therapeutics will require careful consideration of the cellular origins of iPSCs (*Polo et al., 2010*; *Hu et al., 2016*). Although reprogramming of somatic cells to iPSCs involves extensive modification of the epigenetic landscape, the reprogrammed cells can retain an epigenetic memory of the cell type of origin, thus affecting lineage differentiation propensity (*Kim et al., 2010*; *Kim et al., 2011*; *Poetsch et al., 2022*). In addition to donor cell type, key questions over the influence of the health status of the parental somatic cells used for reprogramming remain unresolved. Thus, the present study was designed to determine whether the health status of donor human articular chondrocytes influences the regenerative potential of the derived iPSCs. Using iPSCs generated from healthy and OA chondrocytes, we report that reprogramming efficiency to pluripotency was largely equivalent between the two sources. However, OA-iPSCs showed a significantly reduced capacity for chondrogenic differentiation as compared to AC-iPSCs, indicating that the pathogenic condition of the donor chondrocytes negatively affected the chondrogenic differentiation potential of OA-iPSCs. Our data suggest that reprogramming does not reset the health status of OA articular chondrocytes, but rather supports the existence of a memory of disease in iPSCs derived from OA cartilage.

A plethora of studies over the last 15 years have determined that cells from almost any tissue can be used to generate human iPSCs, which can then be differentiated to a variety of specialized cells. However, human iPSCs generated from disparate cell types have not displayed equivalent capacities for differentiation to specialized cell types (*Kim et al., 2011*). Seminal studies using iPSCs derived from myeloid cells, hematopoietic cells, and insulin-producing beta cells revealed a biased lineage differentiation attributed to residual DNA methylation signatures that influence cell fate commitment. For example, when compared to isogenic non-beta cell-derived iPSCs, beta cell-derived iPSCs maintained an open chromatin structure at key beta cell genes, leading to an increased capacity to efficiently differentiate into insulin-producing cells (*Bar-Nur et al., 2011*). Thus, iPSCs appear to have an epigenetic memory for the tissue of origin. We have previously generated iPSCs from multiple cell types including human skin fibroblasts, umbilical cord blood, and normal healthy chondrocytes using the same reprogramming strategy (*Guzzo et al., 2014*). Using multiple chondrogenic differentiation assays, our earlier findings demonstrated that iPSC derived from chondrocytes showed enhanced matrix formation and chondrogenic gene expression, suggesting that the tissue of origin also impacted the chondrogenic potential of human iPSCs (*Guzzo et al., 2014*) Similarly, a previous report also demonstrated the differential chondrogenic capabilities of iPSCs derived from dermal fibroblasts, peripheral blood mononuclear cells, cord blood mononuclear cells, and OA fibroblast-like synoviocytes (*Rim et al., 2018*).

Although it is now well documented that the tissue of origin can affect the differentiation potential of iPSCs (*Kim et al., 2011*), it is not known whether the health status of same tissue affects the regenerative potential of its derived iPSCs. A combination of genetic and non-genetic factors, including advanced age, mechanical trauma and inappropriate joint loading, and inflammation contribute to the development of OA (*Goldring and Otero, 2011*; *Loeser et al., 2012*; *Martel-Pelletier et al., 2016*). It is well established that human OA articular chondrocytes exhibit phenotypic, functional, and metabolic changes, as well as altered epigenetic patterns (*Izda et al., 2021*). Thus, we speculated that a retained epigenetic memory of iPSCs is not only specific to the tissue of origin but also

to the diseased status. Using pluripotency as a reliable tool, our novel data demonstrated significant differences in the chondrogenic capability of AC- vs. OA-iPSCs. Our data indicate that OA cartilage-derived iPSCs retained functional and molecular characteristics of OA pathogenesis. Although pellet culture used here is a very common method of chondrogenic differentiation of MSCs, it possessed technical limitations of heterogeneity during staining of the chondrogenic-specific matrix in the histological analysis of these macroscopic pellets. Moreover, intensity of safranin-O staining of proteoglycan matrix is uneven from periphery to center which poses another challenge for quantification of matrix deposition in the culture of AC- vs. OA-iMSCs. Additionally, GWAS studies have revealed multiple SNPs in genes involved in OA pathogenesis. We have not yet investigated whether the generated iPSC lines harbor OA-associated sequence variants. Chondrogenic pellets generated using OA-iPSCs showed relatively smaller size and reduced chondrogenic gene expression as compared to that from healthy iPSCs (AC-iPSCs). One of the potential limitations of in vitro differentiation of iMSCs was that the derived chondrocytes express the markers of hypertrophy such as *COL10A1*. Although *COL10A1* expression was observed only with BMP2 stimulation, we did not observe any hypertrophic marker expression when TGFβ3 was used as chondrogenic factor in differentiation medium. Despite these limitations, our comprehensive functional, and transcriptomic analyses support the notion that OA-specific iPSC lines may be useful in vitro tools for studying the underlying molecular, metabolic, and epigenetic mechanisms involved in OA.

Expression of the trio of SOX genes (*SOX9*, *SOX5*, and *SOX6*) was significantly lower in OA-iPSC-derived chondrogenic pellets. The expression of chondrogenic genes under control of SOX9, such as *COL2A1* and *ACAN* was also lower than that AC-iPSC-derived pellets. Since expression of *COL2A1* and *ACAN* is usually lower in OA cartilage, the finding of reduced chondrogenic genes in OA-iPSCs during chondrogenesis suggests the imprints of disease pathogenesis in OA-iPSCs. Similarly, other cartilage matrix genes such as *COMP*, *MATN4*, *PRG4*, and *COL11A2* were lower in OA-iPSC-derived chondrocytes. Interestingly, expression of these genes was also reported to be decreased in OA (*Sofat, 2009*) further suggesting the recapitulation of memory of disease in OA-iPSCs as compared to AC-iPSCs. Additionally, exposure of AC-iMSCs to inflammatory stimulus such as IL1β or TNF-α showed expression of inflammatory cytokine and chemokines such as *CCL2*, *CCL3*, *CXCL3*, and *NOS2*. However, extent of gene expression upon inflammatory stimulation was significantly higher in OA-iMSCs as compared to AC-iMSCs indicating that OA-iPSCs retain memory of disease from the tissue of origin derived from OA cartilage. Based on these findings, human stem cell models of OA using iPSCs may provide the unique opportunity to model OA disease changes, to uncover mechanisms of disease development, and to identify molecular targets for therapeutic intervention.

We next identified how a memory of cartilage pathology in OA is transmitted from the original somatic cells to iMSCs and finally to the chondrocytes. A nonbiased, high-throughput RNA-sequencing approach was used to define the pan-transcriptome changes during iPSC stage-specific differentiation. Our global transcriptome data showed skewed expression of epigenetic regulators, and metabolism-associated molecular pathways in AC- vs. OA-iMSCs, suggesting a transcriptional memory of disease mechanisms in OA-iPSCs. Recent studies showed cellular metabolism as a key driver of cell fate changes which has intrinsic links with epigenetic modifications of chromatin during development, disease progression, and cellular reprogramming (*Wu et al., 2016*). Our data suggest that AC- and OA-iMSCs differ in the expression of a plethora of metabolic genes which finally influence the cells metabolism and thus chondrogenic differentiation. While cell metabolism is closely linked to chondrogenic differentiation, in-depth metabolomic studies are needed to determine how metabolic heterogeneity of AC- and OA-iPSCs impact chondrogenic differentiation and regenerative potential of cartilage tissue. While recent discovery, *Wu et al., 2016* demonstrated the interaction between energy metabolite and epigenetic modifiers, a detailed future investigation warrants to determine how cellular metabolism wired the epigenetic modification and influence the cellular transitions associated with cartilage development.

Although an apparent memory of disease can impact the chondrogenic capabilities of OA-iPSCs, we did not detect differences in stemness genes between AC and OA lines. These data indicate that transcriptome-level differences were notable only upon initial differentiation toward uncommitted mesenchymal progenitors (iMSC stage). We do not know whether the functional and molecular alteration in OA-iMSCs represent a transient or stable phenomenon. Functional studies, coupled with comprehensive analyses of epigenetic landscapes will be necessary to address whether the observed

memory of disease (epigenetic and metabolic) is a stable imprint of the original cellular phenotypes, or could be erased by serial reprogramming. Moreover, does preservation of an epigenetic memory of cartilage disease in iPSCs occur at the DNA methylation level, and if so, what are the OA-associated loci? Further, it is not clear whether memory of disease is a phenomenon observed only at early passages after pluripotency induction or can be attenuated by continuous passaging.

In the present study, we addressed for the first time that differential chondrogenic potential of AC- and OA-iPSCs could be attributed to differences in transcriptome-level changes in the epigenetic modifiers and energy metabolic genes. The expression profile of several chromatin modifiers belonging to the family of histone readers, writers, and erasers such as *FBL*, *PRMT1*, *UBE2E1*, *VRK1*, *PCGF1*, *USP12*, *HMGN3*, *HDAC3*, *HDAC8*, *BRDT*, *ARID2*, and *HMGN3* were significantly different between AC- and OA-iMSCs. Among various epigenetic regulators expressed in AC- vs. OA-iMSCs, the upstream transcription factor analysis identified several candidate factors such as FOXM1, IRF3, FOXP1, and MYBL2. These potential regulators may play a role in response to iMSCs in pathologic or hostile environment under inflammatory stimulus. Since upregulation of FOXM1 is associated with increased response of chondrocytes to inflammation, we anticipate that inhibition of FOXM1 will reduce the pro-inflammatory response and retained regenerative capacity of iMSCs. It is likely that the epigenetic modifications arising in OA chondrocytes occur at multiple levels – DNA methylation, histone modifications, as well as miRNA, and are interconnected. Based on the literature, we speculate that differential DNA methylation at the promoters and regulatory sites of genes that are dysregulated in the parental OA chondrocytes are heritable, and therefore retained in the OA-iPSCs and their derivatives. Our follow-up studies will thoroughly examine DNA promoter methylation status, as well as chromatin organization (loss/gain of active vs. repressive marks) to identify potential regulatory sites that could be targeted to resist pathologic change and retain regenerative capacity. We believe that the generation of OA-iPSCs and the functional and transcriptomic characterization of these novel lines present an important step toward establishing an experimental in vitro model to study disease mechanisms.

In addition, several metabolic genes such as *AOX1*, *OGDHL*, *GATM*, *KMT2D*, *ALDH2*, *GOT1*, *SLC3A2*, and *ECHS1* also showed differential expression pattern between AC- and OA-iMSCs. Several studies previously showed that metabolic genes and metabolites are involved in the regulation of histone acetylation and chromatin modification indicating that importance of chromatin and metabolites in physiological function of the cells (*Jo et al., 2020*; *Schvartzman et al., 2018*). Future studies using genome editing approaches coupled with metabolomics and chromatin mapping approaches will be required to determine the biological roles of these identified chromatin modifiers and metabolic regulators in chondrogenic differentiation of iMSCs. Further correlation of chondrogenic differentiation potential of iMSCs derived from chondrocytes from multiple donors, and with varying grades of OA severity will further help establish the concept of epigenetic memory of disease and determine the influence epigenetic and metabolic imprints on cartilage repair and regenerative medicine.

## Materials and methods

### Key resources table

| Reagent type (species) or resource | Designation | Source or reference | Identifiers | Additional information |
|---|---|---|---|---|
| Biological sample (Human) | AC-iPSCs | Generated at corresponding author lab at UConn Health *Guzzo et al., 2013* | | |
| Biological sample (Human) | OA-iPSC | Generated at corresponding author lab at UConn Health | | |
| Biological sample (Human) | AC-iMSCs | Derived from AC-iPSCs | | |
| Biological sample (Human) | OA-iPSC | Derived from OA-iPSCs | | |
| Antibody | FITC Mouse Anti-Human CD44, Mouse monoclonal | BD-Biosciences | 347943, RRID:AB_400360 | 1:100 for flow cytometry |

*Continued on next page*

*Continued*

| Reagent type (species) or resource | Designation | Source or reference | Identifiers | Additional information |
|---|---|---|---|---|
| Antibody | PE Mouse Anti-Human CD73, Mouse monoclonal | BD-Biosciences | 550257, RRID:AB_393561 | 1:100 for flow cytometry |
| Antibody | FITC Mouse Anti-Human CD90, Mouse monoclonal | BD-Biosciences | 555595, RRID:AB_395969 | 1:100 for flow cytometry |
| Antibody | PE Mouse Anti-Human CD166, Mouse monoclonal | BD-Biosciences | 559263, RRID:AB_397210 | 1:100 for flow cytometry |
| Antibody | FITC Mouse Anti-Human CD105, Mouse monoclonal | BD-Biosciences | 561443, RRID:AB_10714629 | 1:100 for flow cytometry |
| Antibody | FITC Mouse Anti-Human CD31, Mouse monoclonal | BD-Biosciences | 555445, RRID:AB_395838 | 1:100 for flow cytometry |
| Antibody | FITC Mouse Anti-Human CD45, Mouse monoclonal | BD-Biosciences | 347463, RRID:AB_400306 | 1:100 for flow cytometry |
| Antibody | FITC Mouse IgG1, κ Isotype Control, mouse, clonality unknown | BD-Biosciences | 349041, RRID:AB_400397 | 1:100 for flow cytometry |
| Antibody | PE Mouse IgG1, κ Isotype Control, Mouse monoclonal | BD-Biosciences | 555749, RRID:AB_396091 | 1:100 for flow cytometry |
| Commercial assay or kit | High-capacity cDNA Reverse | Transcription Kit Applied Biosystems | 4368814 | |
| Commercial assay or kit | Powerup SYBR green Mix | Thermo Fisher | A25742 | |
| Chemical compound and drugs | Trizol | Thermo Fisher | 15596026 | |
| Chemical compound and drugs | DMEM | Thermo Fisher | 11965092 | |
| Chemical compound and drugs | Recombinant Human FGF-basic | Peprotech | 100-18B | |
| Chemical compound and drugs | Non-Essential Amino Acids Solution | Thermo Fisher | 11140050 | |
| Chemical compound and drugs | HyClone Fetal Bovine Sera Defined | VWR | 16777-006 | |
| Chemical compound and drugs | Pen Strep | Thermo Fisher | 10378016 | |
| Chemical compound and drug | L-Ascorbic acid | Sigma | A4544 | |
| Chemical compound and drug | Glutamax 100X | Gibco | 35050-061 | |
| Chemical compound and drug | Dexamethasone | Sigma | D2915 | |
| Chemical compound and drug | L-Proline | Sigma | P0380 | |
| Chemical compound and drug | Insulin–transferrin–selenium | Thermo Fisher | 41400045 | |
| Software and algorithm | Prism | GraphPad | RRID:SCR_002798 | |
| Software and algorithm | DESeq2 | Bioconductor | DESeq2, RRID:SCR_015687 | |

## iPS cell induction and culture

We have previously described the generation iPSCs reprogramming from human chondrocytes isolated from normal healthy cartilage (AC-iPSCs) (*Guzzo et al., 2014*). These iPSCs were fully reprogrammed and detailed characterization of pluripotency were performed previously using various methods including molecular, cytochemical, cytogenic, and in vitro and in vivo functional analyses (*Guzzo et al., 2014*). Using similar methods, we derived and characterized iPSCs from OA chondrocytes (OA-iPSCs).

**Table 1.** Primer sequences.

| Genes | Forward primer (5'→3') | Reverse primer (5'→3') |
|---|---|---|
| *OCT3/4* (NM_203289) | TGTACTCCTCGGTCCCTTTC | TCCAGGTTTTCTTTCCCTAGC |
| *NANOG* (NM_024865) | CAGTCTGGACACTGGCTGAA | CTCGCTGATTAGGCTCCAAC |
| *KLF4* (NM_004235) | TATGACCCACACTGCCAGAA | TGGGAACTTGACCATGATTG |
| *SOX9* (NM_000346) | AGACAGCCCCCTATCGACTT | CGGCAGGTACTGGTCAAACT |
| *ACAN* (NM_013227) | TCGAGGACAGCGAGGCC | TCGAGGGTGTAGCGTGTAGAGA |
| *COL2A1* (NM_001844) | GGCAATAGCAGGTTCACGTACA | CGATAACAGTCTTGCCCCACTT |
| *COL10A1* (NM_000493) | CAAGGCACCATCTCCAGGAA | AAAGGGTATTTGTGGCAGCATATT |
| *ACTB (NM_001101.5)* | CTC TTC CAG CCT TCC TTC CT | AGCACTGTG TTG GCGTAC AG |

The OA-iPSCs were generated at UConn Health with IRB approval. We procured surgical discards from a 77-year-old female patient undergoing knee joint replacement surgery at our clinic. Chondrocytes were harvested from remaining, OA-affected cartilage at the tibia plateau. OA-derived iPSCs were generated using polycistronic STEMCAA lentiviral vector (as described in our previous publication).

We used three clones from each of the AC-iPSCs (clones #7, #14, and # 15) and OA-iPSC (clones #2, #5, and #8) to ensure that our data are not clone specific. The iPSC colonies were maintained in undifferentiated pluripotent state by culturing the cells under feeder free conditions on 0.1% Geltrex (Peprotech)-coated culture plates. For routine expansion, iPSCs colonies were passaged after reaching 70% confluency using treatment of ReLeSR reagent (StemCell Technologies) and cultured in new 6-well plate using mTeSR plus medium supplemented with 10 µM Y-27,632 Rock inhibitor (Stem-Cell Technologies). Pluripotency of all lines was established by analyzing the expressions of canonical stemness genes (*SOX2*, *NANOG*, *OCT4*, and *KLF4*) using qPCR assay as described previously (***Khan et al., 2017***). Full list of primers is listed in ***Table 1***. We also performed immunofluorescence staining for pluripotency markers in these iPSC colonies using Pluripotent Stem Cell 4-Marker Immunocyto-chemistry Kit (Thermo Fisher Scientific) as per the manufacturer's instruction and fluorescence were imaged using fluorescence microscopy (BioTek Lionheart LX Automated Microscope) as described previously (***Diaz-Hernandez et al., 2020***). ALP staining was also performed for pluripotency charac-terization using TRACP & ALP double-stain Kit (Takara) following the manufacturer's instructions. ALP-positive colonies were imaged using Automated Microscope (BioTek Lionheart LX).

## Derivation of mesenchymal progenitor cells from AC- and OA-iPSCs

Differentiation of iPSCs into chondrocytes requires an intermediate state which we termed as uncom-mitted mesenchymal progenitor cells or MSCs. The differentiation of iPSCs into MSCs was performed using our established direct plating method as described previously (***Guzzo and Drissi, 2015***; ***Guzzo et al., 2013***). Briefly, cell suspensions of iPSC colonies (P15–17) were prepared using accutase treat-ment followed by seeding onto gelatin-coated culture plate using MSC growth medium consisting of DMEM (Dulbecco's Modified Eagle Medium )-high glucose (Gibco), 10% defined fetal bovine serum (FBS; Hyclone), 1% nonessential amino acids, 1× penicillin–streptomycin, and 5 ng/ml rhbFGF (Peprotech). After 2–3 passage onto non-coated plates, the heterogenous cultures acquired the iPSC-MSC-like homogenous, fibroblast-like morphology which was termed as iPSC-derived MSCs (referred as iMSCs). For routine expansion, AC-iPSC- and OA-iPSC-derived MSCs (AC- and OA-iMSCs) were plated at density of 0.3–0.4 × 10$^6$ cells per 100 mm culture dish and maintained in MSC growth media. The characterization of the MSC like feature was performed using gene expression analysis of mesen-chymal genes by qPCR assay as described previously (***Diaz-Hernandez et al., 2020***).

## Flow characterization of mesenchymal progenitor cells (iMSCs)

Immunophenotyping analysis for cell surface markers was performed as defined by the International Society for Cell & Gene Therapy (ISCT) for the minimal criteria of MSCs (***Dominici et al., 2006***). Surface staining of MSCs markers were performed using labeled anti-human antibody against CD73, CD95, CD105, CD44, CD45, CD31, and CD29 using method described previously (***Khan and Poduval,***

*2011a*; *Khan and Poduval, 2012*; *Khan et al., 2011b*). Isotype-matched control (IgG1-PE and IgG2b-FITC) were used for identifying nonspecific fluorescence. Cells were acquired using BD FACSAria using FACS Diva software (Becton–Dickinson). For each analysis, minimum of 20,000 cells was acquired and data were analyzed using FlowJo Software as described previously (*Diaz-Hernandez et al., 2020*).

## Chondrogenic differentiation of iMSCs

We performed chondrogenic differentiation of iMSCs (P18–22) in 3D high-density culture conditions using pellet suspension and micromass adherent method using our established protocol as described previously (*Guzzo et al., 2014*; *Guzzo and Drissi, 2015*; *Guzzo et al., 2013*; *Drissi et al., 2015*). Briefly, for pellet culture, single-cell suspension of AC- and OA-iMSCs culture was performed using 0.25% trypsin–EDTA (Ethylenediaminetetraacetic acid) and $0.5 \times 10^6$ cells were placed in 15-ml poly-propylene tubes and centrifuged at $300 \times g$ for 5 min to pellet the cells, and finally cultured in MSC growth medium in $CO_2$ incubator at 37°C and 5% $CO_2$ for 1 day. Twenty-four hours after pellet formation, the culture media was replaced with chondrogenic media consisting of DMEM-high glucose media (Gibco), 1% ITS (Insulin-Transferrin-Selenium) premix, 40 µg/ml L-proline, 1 mM sodium pyruvate, 1× nonessential amino acids, 1× Glutamax, 50 µg/ml ascorbic acid 2-phosphate, and 0.1 µM dexamethasone, 1× penicillin/streptomycin, and human recombinant BMP-2 (100 ng/ml, Peprotech) (*Guzzo and Drissi, 2015*). Chondrogenic media and growth factor were changed every other day until the end of 21 days of chondrogenic differentiation. Cell pellets were harvested at 0, 7, 14, and 21 days of differentiation and analyzed for gene expression using SYBRGreen qPCR assay.

Chondrogenic differentiation was also performed using adherent micromass method (*Guzzo and Drissi, 2015*). The micromass of AC- and OA-iMSCs was prepared by culturing the cells in high density ($25 \times 10^4$ cells per 10 µl drop) in 6-well culture plates. Immediately after seeding the micromasses, MSC growth medium was carefully added dropwise from the edges of the plate to prevent dehydration of micromass. These micromasses were incubated for 4–6 hr at 37°C in 5% $CO_2$, and then supplemented with 2 ml of MSC growth medium and cultured for 24 hr. Then MSC growth media was replaced with chondrogenic media and differentiation was continued for 21 days. Micromass culture was harvested at different days of chondrogenic differentiation and processed for either RNA isolation or fixed with formalin for Alcian blue staining. Formalin-fixed micromass cultures were stained with 1% Alcian blue in acetic acid, pH 2.5 and proteoglycans levels were measured by imaging the blue colonies using automated microscope (BioTek Lionheart LX).

## Osteogenic and adipogenic differentiation of iMSCs

To establish the multilineage potential of the iMSCs, we next assessed the ability of AC- and OA-iMSCs to differentiate into osteogenic and adipogenic lineages in vitro. Osteogenesis of iMSCs was induced by culturing 10,000 cells per well of 24-well plate using osteogenic medium consisting of DMEM supplemented with 1 mM sodium pyruvate, 0.1 µM dexamethasone, 50 µg/ml ascorbic acid 2-phosphate, 10 mM β-glycerophosphate, 10% FBS and 1× penicillin/streptomycin for 21 days. At end of 21 days culture, cells were fixed with formalin and stained for Alizarin red solution to visualize calcium deposits as described previously (*Guzzo et al., 2014*; *Guzzo and Drissi, 2015*; *Guzzo et al., 2013*).

Additionally, to induce adipogenesis, the iMSCs were seeded at 10,000 cells per well in 24-well plate and cultured for 21 days in presence of adipogenic media consisting of DMEM-high glucose supplemented with 10% FBS, 1 mM sodium pyruvate, 1 µM dexamethasone, 10 µg/ml insulin, 0.5 mM isobutylmethylxanthine, 200 µM indomethacin, and 1× penicillin/streptomycin. Adipogenesis was measured by Oil Red O staining of formalin-fixed cells for detection of lipid accumulation as described previously (*Guzzo et al., 2014*; *Guzzo and Drissi, 2015*; *Guzzo et al., 2013*).

## RNA-sequencing of iMSCs during chondrogenic differentiation

To examine the transcriptional changes during chondrogenic differentiation using the pellet method, we performed RNA-sequencing of AC- and OA-iMSCs during the course of differentiation process. Pellets from both AC- and OA-iMSCs were isolated at 0, 7, 14, and 21 days of chondrogenic differentiation and total RNA was isolated using miRNeasy Kits. On-Column DNase digestion was performed to remove genomic DNA contamination. RNA quality was checked using Nanodrop and the RNA integrity was determined by Agilent 2200 Bioanalyzer, and the RNA integrity number values were >7 for

all samples. Libraries were prepared from 250 ng RNA using TruSeq Stranded Total RNA Sample Prep Kit (Illumina) using the Poly A enrichment method. Sequencing was carried out using the NovaSeq PE 150 system (Novogene UC Davis Sequencing Center, Novogene Corporation Inc). Raw data were exported in FASTQ (fq) format and quality control was performed for error rate and GC content distribution, and data filtering was performed to remove low-quality reads or reads with adaptors. The clean reads were mapped to human reference genome (GRCh38) and differential gene expression (DEG) analysis was performed using DESeq2 method and pairwise gene expression levels were calculated using RPKM (read per kilobase of transcript sequence per millions base pairs sequenced) value. FC (fold change) in gene expression was performed on filtered datasets using normalized signal values.

## Differential gene expression analysis of RNA-seq data

DEGs were identified using DESeq2 in R Bioconductor (*Love et al., 2014*). Log FC represented the fold change of gene expression, and $p < 0.05$ and $log_2$ FC >2 was set for statistically significant DEGs. Multiple correction testing was performed using false discovery rate (FDR). The DEGs between AC- and OA-iMSCs at day 21 were visualized using heatmap, volcano plot, and principal component analysis using R-Bioconductor package as described previously (*Diaz-Hernandez et al., 2020*; *Fernandes et al., 2020*; *Khan et al., 2020a*; *Khan et al., 2020b*). Molecular pathways enriched in DEGs were performed using GO (Gene Ontology) and KEGG pathways analysis using STRING (v11.0) (*Szklarczyk et al., 2017*). The enrichment of top GO terms based on FDR corrected p value was visualized by dot plot analysis as described previously (*Khan et al., 2020a*; *Khan et al., 2020b*). *X*-axis in the dot plot represents 'gene term ratio', which was calculated by ratio of gene numbers enriched in a particular GO term to all the gene numbers annotated in that GO term.

We also performed differential gene expression analysis between healthy and OA chondrocytes by analyzing the publicly available RNA-seq datasets. The raw data were downloaded from healthy and OA cartilage tissues (GSE114007) available from the NCBI-GEO database (*Fisch et al., 2018*). DEGs were identified using DESeq2 in R Bioconductor as described above. The heatmap for mRNA expression profiling of selected genes was generated by R package of pheatmap as described previously (*Khan et al., 2020b*).

## Interaction network analysis of DEGs between AC- and OA-iMSCs

To identify the interactions among top DEGs between AC- and OA-iMSCs during chondrogenic differentiation, we performed interaction network analyses using STRING database (v11.0) using a stringent criterion with a combined score of >0.7 showing most significant interactions (*Szklarczyk et al., 2017*). Network clusters were identified using connectivity degree and hub proteins were identified as node showing maximum clustering score in the interaction network. The interaction network was visualized by the Cytoscape (v3.9.0), a bioinformatics package for biological network visualization and data integration (*Otasek et al., 2019*) as described previously (*Khan et al., 2020a*; *Khan et al., 2020b*). Significant clusters in the interaction network were analyzed by subnetwork analysis using the Molecular Complex Detection Algorithm (MCODE) plugin (v1.5.1) in Cytoscape (*Saito et al., 2012*). Enrichment of molecular pathways in identified network cluster was analyzed using ClueGO analysis in Cytoscape (*Bindea et al., 2009*). The genes identified in metabolic and epigenetic regulator pathways in network clusters were also analyzed for differential expression analysis between AC- and OA-iMSCs and visualized by heatmap analysis.

## Statistics

Data are expressed as mean ± standard error of the mean of at least three independent experiments. All experiments represent biological replicates and were repeated at least three times, unless otherwise stated. Technical replicates are repeat tests of the same value, that is, testing same samples in triplicate for qPCR. Biological replicates are samples derived from separate sources, such as different clones of iPSCs and iMSCs. Statistical comparisons between two groups (AC vs. OA) were performed using a two-tailed Student's *t*-test for comparing two groups using GraphPad Prism. One-way analysis of variance followed by Tukey's test multiple comparisons test for greater than two groups Significance was denoted at $p < 0.05$.

## Acknowledgements

This work was supported by funds from Veteran Affairs and Emory University School of Medicine. This research was funded by Georgia CTSA/REM Pilot Project 00080502 to HD and LJM, Veteran Affairs CaReAP Award (I01-BX004878) to HD, and Connecticut Innovation Stem Cell Fund Seed Grants (#13-SCA-UCHC-11, #10SCA36) to RG.

## Additional information

### Funding

| Funder | Grant reference number | Author |
| --- | --- | --- |
| Veterans Administration Medical Center | CaReAP Award I01-BX004878 | Hicham Drissi |
| Georgia Clinical and Translational Science Alliance | REM Pilot Project 00080502 | Luke J Mortensen Hicham Drissi |
| Connecticut Innovations | Stem Cell Fund Seed Grants #13-SCA-UCHC-11 | Rosa M Guzzo |
| Connecticut Innovations | Stem Cell Fund Seed Grants #0SCA36 | Rosa M Guzzo |

The funders had no role in study design, data collection, and interpretation, or the decision to submit the work for publication.

### Author contributions

Nazir M Khan, Conceptualization, Data curation, Software, Formal analysis, Investigation, Visualization, Methodology, Writing – original draft, Writing – review and editing; Martha Elena Diaz-Hernandez, Data curation, Validation, Investigation, Visualization, Methodology; Samir Chihab, Data curation, Validation, Investigation, Methodology; Priyanka Priyadarshani, Investigation, Methodology; Pallavi Bhattaram, Formal analysis, Visualization, Writing – review and editing; Luke J Mortensen, Resources, Funding acquisition, Project administration, Writing – review and editing; Rosa M Guzzo, Data curation, Funding acquisition, Methodology, Writing – review and editing; Hicham Drissi, Conceptualization, Resources, Software, Supervision, Funding acquisition, Project administration, Writing – review and editing

### Author ORCIDs

Nazir M Khan http://orcid.org/0000-0002-1637-7641
Luke J Mortensen http://orcid.org/0000-0002-4331-4099
Hicham Drissi http://orcid.org/0000-0002-3322-281X

### Decision letter and Author response

Decision letter https://doi.org/10.7554/eLife.83138.sa1
Author response https://doi.org/10.7554/eLife.83138.sa2

## Additional files

### Supplementary files

• MDAR checklist

### Data availability

All raw data have been made available as source data files within the manuscript. The sequencing datasets are available via the Gene Expression Omnibus (GEO) under the accession number GSE 214987.

The following dataset was generated:

| Author(s) | Year | Dataset title | Dataset URL | Database and Identifier |
|---|---|---|---|---|
| Khan NM, Drissi H | 2023 | RNA-seq during chondrocyte differentiation of iMSCs derived from iPSCs of healthy (AC-iPSCs) and OA chondrocytes (OA-iPSCs) | https://www.ncbi.nlm.nih.gov/geo/query/acc.cgi?acc=GSE214987 | NCBI Gene Expression Omnibus, GSE214987 |

The following previously published dataset was used:

| Author(s) | Year | Dataset title | Dataset URL | Database and Identifier |
|---|---|---|---|---|
| Fisch KM, Gamini R, Alvarez-Garcia O, Akagi R, Saito M, Muramatsu Y, Sasho T, Ai Su, Lotz MK | 2018 | Identification of transcription factors responsible for dysregulated networks in human osteoarthritis cartilage by global gene expression analysis | https://www.ncbi.nlm.nih.gov/geo/query/acc.cgi?acc=GSE114007 | NCBI Gene Expression Omnibus, GSE114007 |

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
