## [Editor Report]

This manuscript demonstrates that iPSCs retain the molecular transcriptional signature associated with a healthy or osteoarthritic (OA) state, depending upon the origin of donor cells. Using iPSCs derived from healthy (AC-iPSCs) or OA cartilage, the data show that epigenetic and metabolism-specific transcriptional signals affect the subsequent differentiation of iPSCs to chondrocytes. These findings significantly contribute to the field of regenerative medicine and pave the way to further design new approaches to control and regulate the differentiation of iPSCs to desired cell types.

---

## [Decision Letter]

**Decision letter after peer review:**

Thank you for submitting your article "Differential chondrogenic differentiation between iPSC-derived from healthy and OA cartilage is associated with changes in epigenetic and metabolic transcriptomic signatures" for consideration by *eLife*. Your article has been reviewed by 3 peer reviewers, one of whom is a member of our Board of Reviewing Editors, and the evaluation has been overseen by Mone Zaidi as the Senior Editor. The following individual involved in the review of your submission has agreed to reveal their identity: Gaurav Swarnkar (Reviewer #3).

Essential revisions:

1. Do epigenetically unmodified AC-iPSCs acquire an OA-iPSC-like epigenetic and metabolic marks and diminished regenerative capacity when exposed to inflammatory/pathologic conditions? Based on your findings, can you elaborate on potential epigenetic modifications that can be introduced in AC-iPSCs to resist pathologic change and retain their regenerative capacity in a hostile environment

2. In all figure legends, although P values are indicated, please indicate the number of independent replicates. Also, what number of replicates do images represent?

3. Please elaborate on potential limitations encountered in this research, including inherent in vitro differentiation or technical limitations

4. References 13, 14, 16, 17, etc need to be updated with doi or pubmed ID. right now they have "electronic" written in brackets, probably mis-formatting

5. Figure 5 – gene network. it would be useful if the names of the metabolic and epigenetic processes would have a bigger font size and bold color or black color. They are very hard to read.

*Reviewer #1 (Recommendations for the authors):*

This is a comprehensive and well-presented characterization of healthy and OA-iPSC with immediate relevance to regenerative medicine. The findings of this research will inform better design of iPSCs for cartilage regeneration and advance this area of research. The experiments are well-designed, unbiased, and comprehensive. There were only a few issues that require the authors' attention.

1. Do epigenetically unmodified AC-iPSCs acquire an OA-iPSC-like epigenetic and metabolic marks and diminished regenerative capacity when exposed to inflammatory/pathologic conditions? Based on your findings, can you elaborate on potential epigenetic modifications that can be introduced in AC-iPSCs to resist pathologic change and retain their regenerative capacity in a hostile environment

2. In all figure legends, although P values are indicated, please indicate the number of independent replicates. Also, what number of replicates do images represent?

3. Please elaborate on potential limitations encountered in this research, including inherent in vitro differentiation or technical limitations

*Reviewer #2 (Recommendations for the authors):*

The manuscript is appropriate for publication. Several comments to be addressed in this manuscript, or future manuscripts, include.

- It would be interesting to examine Prg4 expression in both healthy and OA iPSC cultures, if possible.

- It would also be interesting to examine cell-surface markers for three subsets of skeletal stem and progenitor populations (integrin α V (CD51)+Thy-1 (CD90)−); mouse skeletal stem cells (mSSCs) (CD105−CD200+), pre-bone, cartilage and stromal progenitors (pre-BCSPs) (CD105−CD200−), and bone, cartilage and stromal progenitors (BCSPs) (CD105+). (Chan, C. K. et al. Identification and specification of the mouse skeletal stem cell. Cell 160, 285-298 (2015).)

- It may be of future interest to measure some of the important epigenetic and metabolic marks in the OA patient-derived and healthy donor-derived iPSC lines as well as to know if the iPS cells also have the same characteristic of mesenchymal stem cells from OA patients.

*Reviewer #3 (Recommendations for the authors):*

The findings from this research significantly contribute towards the field of regenerative medicine and cartilage biology. The identification of a new molecular mechanism regulating the discrepancy associated with healthy vs diseased donor cells and differential outcomes of iPSC differentiation will lead to designing a better strategy for cartilage repair. The manuscript is well-written, and the experimental design and results are clearly described. There were only a few issues that require attention.

1. In Figures 1C and 2B, since the comparison is being done between AS vs OA-iPSCs/MSC, it is better to plot them in the same graph (something like Graph 3C).

2. It may be helpful if graphs can be presented in a simpler way where AC vs OA can data graph and each clone represents individual biological replicates (Bar graph with individual data points for biological replicates). In other words, the data from 3 different clones can be presented in 1 bar in place of 3.

3. Were the images taken at different magnifications as the scale bars seem to be different in the images (Figure 1C)?

4. Differential chondrogenic differentiation of AC-iMSCs and OA-MSCs was also confirmed with qPCR analysis. Was any qPCR analysis done to support OB and adipocytic differentiation phenotype?

---

## [Author Response]

Essential revisions:1. Do epigenetically unmodified AC-iPSCs acquire an OA-iPSC-like epigenetic and metabolic marks and diminished regenerative capacity when exposed to inflammatory/pathologic conditions?

We have performed additional analysis addressing this question. When exposed to either IL1β, or TNF-α, AC-iPSCs derived iMSCs (AC-iMSCs) showed increased expression of inflammatory cytokines and chemokines (*CCL3, CXCL3, and PTGS2)* as compared to vehicle-treated controls. Interestingly, inflammatory stimulation resulted in significantly higher levels of induction for markers of inflammation in OA-iMSCs as compared to AC-iMSCs. These new data further support our premise that OA-iPSCs retain a memory of disease from the tissue of origin derived from OA cartilage (Supplementary Figure 3).

Based on your findings, can you elaborate on potential epigenetic modifications that can be introduced in AC-iPSCs to resist pathologic change and retain their regenerative capacity in a hostile environment

We believe that the generation of OA-iPSCs and the functional and transcriptomic characterization of these novel lines present an important step towards establishing an experimental in vitro model to study disease mechanisms. Further evaluation of the epigenetic landscapes between OA-iPSCs and AC-iPSCs may reveal specific changes in the methylome or histone modifications that can be targeted to correct the skewing of OA-iMSCs to provide an equal chondrogenic or immunomodulatory potentials to that of healthy cartilage derived iMSCs. Among various epigenetic marks expressed in AC- vs OA-iMSCs, the upstream transcription factor analysis identified several candidate regulators such as FOXM1, IRF3, FOXP1 and MYBL2 (Supplementary Figure 4). We envision a multitude of follow-up studies aimed at thoroughly examining DNA promoter methylation status, as well as chromatin organization (loss/gain of active versus repressive marks) to identify potential regulatory sites that could be targeted to resist pathologic change and retain regenerative capacity. Our future investigations may also focus on modifying specific metabolic pathways that can normalize chromatin landscapes between these two cell types. We raise these points in the discussion acknowledging that the epigenetic modifications that occur in OA chondrocytes are likely involving a multitude of events such as DNA methylation, histone modifications, as well as miRNA, and which may be inter-connected.

2. In all figure legends, although P values are indicated, please indicate the number of independent replicates. Also, what number of replicates do images represent?

We thank the Reviewer for pointing this out. We now indicate the number of replicates in each figure of the revised manuscript. We also report the replicate number for each representative image. A minimum of 3 biological samples were used in each experiment.

3. Please elaborate on potential limitations encountered in this research, including inherent in vitro differentiation or technical limitations

We derived iMSC-like populations to circumvent heterogeneous differentiation outcomes from naïve iPSCs. Relative to other published approaches, we have established a simple, yet highly efficient protocol for iMSC differentiation to the chondrogenic lineage. However, we acknowledge that the challenges associated with generating a homogenous population of non-hypertrophic chondrocytes from iPSCs persist. It is notable that *COL10A1* expression was observed only with BMP2 stimulation in chondrogenic differentiation media and was not observed when TGFβ3 was used to induce chondrogenic differentiation in our cultures. Other groups have reported a highly stringent approach for human iPSCdifferentiation into chondrocytes with an articular-like phenotype, with low COL10A1 matrix deposition using a complex cocktail of growth factors (WNT, activin-A/Nodal, TGF-b, FGF, PDGF, BMP2). In next steps, we may be applying these strategies to evaluate differential propensity of AC-iPSCs vs OA-iPSC for differentiation to an articular-like chondrocyte phenotype.

While both AC- and OA-iMSCs have developmental plasticity, their chondrogenic differentiation pattern and differentiation efficiency was different from each other’s which pose technical limitation during neck-to-neck comparison. Although pellet culture used here is a very common method of chondrogenic differentiation of MSCs, it possessed technical limitations of heterogeneity during staining of the chondrogenic-specific matrix in the histological analysis of these macroscopic pellets. Moreover, intensity of safranin-O staining of proteoglycan matrix is uneven from periphery to center which poses another challenge for quantification of matrix deposition in the culture of AC- vs OA-iMSCs. Additionally, GWAS studies have revealed multiple SNPs in genes involved in OA pathogenesis. We have not yet investigated whether the generated iPSC lines harbor OA-associated sequence variants.

Despite these limitations, our comprehensive functional, and transcriptomic analyses support the notion that OA-specific iPSC lines may be useful in vitro tools for studying the underlying molecular, metabolic and epigenetic mechanisms involved in OA.

4. References 13, 14, 16, 17, etc need to be updated with doi or pubmed ID. right now they have "electronic" written in brackets, probably mis-formatting

We apologize for this oversight and have corrected the references.

5. Figure 5 – gene network. it would be useful if the names of the metabolic and epigenetic processes would have a bigger font size and bold color or black color. They are very hard to read.

Thank you for this valid critique. Per the Reviewer’s suggestion, we modified the font size and color in the gene network Figure 5 A, B in the revised manuscript.

Reviewer #1 (Recommendations for the authors):This is a comprehensive and well-presented characterization of healthy and OA-iPSC with immediate relevance to regenerative medicine. The findings of this research will inform better design of iPSCs for cartilage regeneration and advance this area of research. The experiments are well-designed, unbiased, and comprehensive. There were only a few issues that require the authors' attention.1. Do epigenetically unmodified AC-iPSCs acquire an OA-iPSC-like epigenetic and metabolic marks and diminished regenerative capacity when exposed to inflammatory/pathologic conditions? Based on your findings, can you elaborate on potential epigenetic modifications that can be introduced in AC-iPSCs to resist pathologic change and retain their regenerative capacity in a hostile environment2. In all figure legends, although P values are indicated, please indicate the number of independent replicates. Also, what number of replicates do images represent?3. Please elaborate on potential limitations encountered in this research, including inherent in vitro differentiation or technical limitations

Please see the responses above in ‘Essential Revision section’

Reviewer #2 (Recommendations for the authors):The manuscript is appropriate for publication. Several comments to be addressed in this manuscript, or future manuscripts, include.- It would be interesting to examine Prg4 expression in both healthy and OA iPSC cultures, if possible.

Following the Reviewer’s suggestion, we analyzed the expression of the hallmark marker of articular chondrocytes, PRG4 (lubricin) in healthy versus OA-iMSCs over a time course of chondrogenic differentiation (days 7, 14, 21). Interestingly, our quantitative PCR analyses revealed significantly higher levels of PRG4 transcripts in AC-iMSCs as compared to OA-iMSCs at all time points analyzed. This new finding is consistent with the elevated expressions of gene markers of immature chondrocytes (*SOX9, COL2A1*, *ACAN),* suggesting that iPSC derived from healthy chondrocytes have a significantly higher chondrogenic potential as compared to OA-iPSC. These data also suggest increased differentiation of chondrocytes with articular-like chondrocyte phenotype in AC-iMSCs as compared to OAiMSCs. These new data appear in Figure 3C of the revised manuscript.

- It would also be interesting to examine cell-surface markers for three subsets of skeletal stem and progenitor populations (integrin α V (CD51)+Thy-1 (CD90)−); mouse skeletal stem cells (mSSCs) (CD105−CD200+), pre-bone, cartilage and stromal progenitors (pre-BCSPs) (CD105−CD200−), and bone, cartilage and stromal progenitors (BCSPs) (CD105+). (Chan, C. K. et al. Identification and specification of the mouse skeletal stem cell. Cell 160, 285-298 (2015).)

We thank the Reviewer for this suggestion. Identification of skeletal stem and progenitor populations require an in-depth characterization of these iPSC lines including multiple assays such as flow characterization by FACS analysis, colony formation assays and expansion assay in local microenvironment etc. Further, studies from mice have indicated that the SSC phenotype does not stably persist in cell culture, therefore, we believe that in order to appropriately define differences in the progenitor/stem cell pools we need to perform in depth in vitro and in vivo lineage tracing assessments to identify the percent population of these SSCs in AC vs OA-iMSCs which we will thoroughly examine in our follow-up study.

- It may be of future interest to measure some of the important epigenetic and metabolic marks in the OA patient-derived and healthy donor-derived iPSC lines as well as to know if the iPS cells also have the same characteristic of mesenchymal stem cells from OA patients.

We agree with the Reviewer that a comprehensive analysis to identify changes in epigenetic marks, as well as the metabolic marks between OA- versus healthy cartilage derived iPSCs will be informative towards establishing a memory of disease status. Comparative analysis of genome-wide chromatin accessibility (ATACseq), DNA methylation, and untargeted metabolomics between OA-iPSCs and healthy donor derived iPSCs will identify epigenetic and metabolic changes that contribute to reduced chondrogenic differentiation potential in OA-iPSCs, and will be pursued in follow-up studies.

Reviewer #3 (Recommendations for the authors):The findings from this research significantly contribute towards the field of regenerative medicine and cartilage biology. The identification of a new molecular mechanism regulating the discrepancy associated with healthy vs diseased donor cells and differential outcomes of iPSC differentiation will lead to designing a better strategy for cartilage repair. The manuscript is well-written, and the experimental design and results are clearly described. There were only a few issues that require attention.1. In Figures 1C and 2B, since the comparison is being done between AS vs OA-iPSCs/MSC, it is better to plot them in the same graph (something like Graph 3C).

We agree with the Reviewer, and have revised the format of the graphs in accordance with their suggestion.

2. It may be helpful if graphs can be presented in a simpler way where AC vs OA can data graph and each clone represents individual biological replicates (Bar graph with individual data points for biological replicates). In other words, the data from 3 different clones can be presented in 1 bar in place of 3.

Following the Reviewer’s suggestion, we have revised the graph format to represent the 3 different clones in 1 bar.

3. Were the images taken at different magnifications as the scale bars seem to be different in the images (Figure 1C)?

All the images were taken at same 4X magnification.

4. Differential chondrogenic differentiation of AC-iMSCs and OA-MSCs was also confirmed with qPCR analysis. Was any qPCR analysis done to support OB and adipocytic differentiation phenotype?

We thank the Reviewer for this suggestion. We performed gene expression analysis for the markers of osteoblast differentiation in AC- vs OA-iMSCs over a time course of osteoblast differentiation (days 7, 14, 21). Our quantitative PCR analyses revealed that transcripts levels of osteogenic genes such *RUNX2, OSX, COL1A1* was higher at 7 days in comparison to undifferentiated iMSCs levels (day 0). The transcript level of these genes was further increased to day 14 and day 21 of osteogenic differentiation. Interestingly, the expression of these osteogenic genes in AC-iMSCs was not statistically significant when compared to OA-iMSCs at all time points analyzed. This new finding is consistent with the Alizarin red staining showing that AC- and OA-iMSCs exhibit similarities in osteogenic differentiation. These data were shown in Supplementary Figure 1C of revised manuscript.